# Intrinsic dichroism in amorphous and crystalline solids with helical light

Ashish Jain [1,2] ✉, Jean-Luc Bégin [1,2] ✉, Paul Corkum[1], Ebrahim Karimi [1], Thomas Brabec[1] & Ravi Bhardwaj [1] ✉

Amorphous solids do not exhibit long-range order due to the disordered arrangement of atoms. They lack translational and rotational symmetry on a macroscopic scale and are therefore isotropic. As a result, differential absorption of polarized light, called dichroism, is not known to exist in amorphous solids. Using helical light beams that carry orbital angular momentum as a probe, we demonstrate that dichroism is intrinsic to both amorphous and crystalline solids. We show that in the nonlinear regime, helical dichroism is responsive to the short-range order and its origin is explained in terms of interband multiphoton assisted tunneling. We also demonstrate that the helical dichroism signal is sensitive to chirality and its strength can be controlled and tuned using a superposition of OAM and Gaussian beams. Our research challenges the conventional knowledge that dichroism does not exist in amorphous solids and enables to manipulate the optical properties of solids.

Light as a tool to probe the structure and symmetry of solids is limited due to the fact that the wavelengths of photons (UV–visible–IR) are significantly longer than the interatomic distances. In spite of this limitation, some degree of information on symmetry and phonon modes can be obtained from light scattering techniques such as Raman spectroscopy[1,2]. In crystalline solids (c-solids), periodic arrangement of atoms leads to long-range order that gives rise to anisotropy in which the physical properties depend on the crystal orientation. Such an anisotropy in absorption and emission of light via high harmonic generation has been exploited to map the crystal symmetry[3,4]. In chiral c-solids, material handedness (non-super-imposable mirror images), in principle, leads to differential absorption of right- and left-circularly polarized light, known as circular dichroism (CD). However, the chiroptical signal is often influenced by linear birefringence and inherent macroscopic anisotropies present in c-solids[5]. As a result, sophisticated techniques such as single-crystal X-ray diffraction[6], scanning and transmission electron microscope-based methods[7] are often utilized to differentiate chiral crystals but are limited by the need for large single crystal of high purity.

In contrast, amorphous solids (a-solids) do not have long-range order due to random arrangement of atoms and are isotropic. A consequence of the lack of symmetry in a-solids is that they do not exhibit CD. However, a-solids are characterized by short- and medium-range order in which the spatial variation of parameters (such as interatomic distances and angles between neighboring atoms) provides average structural information and the degree to which the short and medium-range order is conserved[8,9]. Studying short- and medium-range order of a-solids is an active field of research[9,10] because it can lead to some phenomena that are typically observed in c-solids. With recent technological advances, it is now possible to directly observe the local atomic structure of disordered solids[11,12]. The presence of short- and medium-range order in a-solids is in fact responsible for delocalized energy valence and conduction band states, whereas the long-range disorder causes the formation of bond tail energy states between energy bands[13]. In amorphous glass, the short to medium-range order extends up to 20 Å[9,14–16].

Most optical techniques to study solids, to date, relied on using polarized Gaussian beams (carrying spin angular momentum $0, \pm 1\hbar$) with the interaction described using the dipole dominant molecular transitions. Such transitions are generally independent of the phase associated with the wavefront of the incident beam. However, light can also carry orbital angular momentum (OAM), $\pm l\hbar$, associated with

[1]Nexus for Quantum Technologies, Department of Physics, University of Ottawa, Ottawa, ON K1N 6N5, Canada. [2]These authors contributed equally: Ashish Jain, Jean-Luc Bégin. ✉e-mail: ajain067@uottawa.ca; jbegi038@uottawa.ca; ravi.bhardwaj@uottawa.ca

dynamical rotation of wavefront structure. The handedness of these beams is characterized by the twisting of the wavefront undergoing $l$ intertwined rotations in one wavelength[17–19]. The intensity profile of such light is characterized by a phase singularity (optical vortex) and hence a null intensity region at the center of the beam. These vortex beams are also referred as helical light beams[19–21] where the helicity is associated with the helical wavefront structure which is chiral in nature. This is analogous to the chiral structure of the electric field associated with circular polarized light.

The theoretical unboundedness of OAM value generated significant interest in using helical light beams as a chiral probe to study light-matter interaction. Early studies on differential absorption of helical light beams focused mostly on the linear absorption involving coupling of electric–magnetic dipole (E1M1) term, which was known to be responsible for CD. These studies were not successful in demonstrating the efficacy of using OAM as a chiral reagent. Recently, Helical Dichroism (HD), defined as differential absorption of left- and right-handed helical light, was demonstrated in nanoparticle aggregates[22], chiral metasurfaces[23], powdered molecular media[24], and achiral/chiral liquids in nonlinear regime[25]. Theoretically, several recent studies proposed the phased-based dichroism effect originates from the higher-order transition moments, which were shown to be non-vanishing only in the medium with some degree of orientational order[26,27]. Traditionally, the origin of such effects can be understood in terms of parity-time (PT) symmetry arguments. Electric dipole transition moment E1 has odd spatial parity, magnetic dipole moment M1 and electric quadrupole moment E2 have even parity. Hence, the E1M1 and E1E2 coupling terms represent a time-even space-odd pseudoscalars which changes sign on spatial inversion indicating a broken symmetry. However, symmetry arguments can only provide qualitative insight into the existence of dichroism in crystals and does not predict its presence in amorphous solids. Therefore, comprehensive models are required for quantitative understanding of the behavior of HD that takes into account the interband transition dynamics in matter and spatial structure of the helical light beams.

In this article, we first demonstrate existence of intrinsic HD in amorphous solids using asymmetrical helical light beams. Such an effect is typically not expected in amorphous materials due to the disorderly state of the medium. We define differential absorption of left- and right-helical light for the same material as HD (Type I). Therefore, it is a beam-dominated property and can also be observed in achiral and chiral crystalline solids. In case of amorphous and achiral solids, HD (Type I) is not considered as true chirality as it does not involve chiral light and chiral matter interaction. Second, our technique demonstrates higher efficiency in probing chiral solids compared to conventional solid-state-based optical techniques. This was achieved in terms of HD (Type II), defined as difference of absorption between the left- and right-handed chiral solid for a specific-handed helical light. It is a material-dominated property and its definition requires both the material and light to be chiral entities. Third, we show that helical dichroism is tunable and can be precisely controlled by (i) superposition of OAM and Gaussian beams (ii) varying the $l$-value, and (iii) displacement of phase singularity in the beam. These features combined with the bandgap dependence of HD set our technique apart from any other existing chiroptical methods. Finally, we model HD by considering electron transitions via multiphoton-assisted tunneling (MPAT). This process ensures that electron displacement remains within the short- to medium-range order in solids, allowing us to effectively probe intrinsic dichroism.

## Results

We studied HD in both amorphous and crystalline solids by measuring the absorption of loosely focused femtosecond helical light pulses (Supplementary Section 1 for experimental setup). We produced helical light beams carrying OAM using a birefringent plate, called q-

plate, in which the incident Gaussian beam acquires an OAM that is two times the topological charge, $q$[21,28]. To disentangle the effects of spin and orbital angular momentum on helical dichroism we used linearly polarized helical light beams. In addition, asymmetric helical light beams were produced by displacing the phase singularity in the OAM beam by translating the q-plate in a plane perpendicular to the incident beam. Additional experimental details are provided in "Methods" ("Differential absorption measurements", "Generation of OAM beams", and "Displacement of singularity") and Supplementary (Sections 5–7).

### Achiral crystalline solids

Helical dichroism, HD (Type I)$=A(+l, s) − A(−l, s)$, defined as differential absorption of linearly polarized ($s = 0$) left- and right-helical light ($l = \pm 3$) in the same material, is shown in Fig. 1a for a MgO achiral crystal as a function of the position of the singularity in the OAM beam. The inset shows normalized transmission of a single left- and right-handed helical light pulse as a function of peak laser fluence (Supplementary Section 4) when the singularity was displaced by 900 nm. Each curve in the inset is an average of three independent measurements, and the color band represents the statistical standard error. Differences in the transmission of the two helicities start to appear at a fluence of ~0.8 J/cm$^2$ from the onset of nonlinear absorption and persist over a broad range of peak laser fluences. For each position of the singularity, HD (Type I) was obtained by averaging transmission (shown in the inset) over a fluence range from the onset of nonlinear absorption to 5 J/cm$^2$.

Dichroism does not exist in achiral solids for a circularly polarized Gaussian beam and also for a linearly polarized symmetric OAM beam ($\delta = 0 \pm \Delta$ in Fig. 1a). However, when the singularity in the OAM beam is displaced from the center, the material exhibits helical dichroism. HD (Type I) signal increases with the displacement, reaches a maximum around ±1500 nm and displays a sinusoidal behavior with a change in sign as the singularity traverses the zero-position (Supplementary Section 7). This can be understood in terms of parity-time symmetry argument. HD (Type I) is odd under PT resulting in a change in sign with respect to the $\delta$ position. In the experiment, alignment of the singularity at the center of the OAM beam can only be defined before the objective, which translates to an uncertainty $\Delta$~±100 nm at the focus (see "Displacement of singularity" for calibration). HD (Type I) signal decreases at large displacement of singularity, as expected, since the intensity profile of asymmetric OAM beam starts to resemble a Gaussian beam.

Orientation-dependent nonlinear absorption of helical light pulses in MgO crystal is shown in Fig. 1b–d for three different displacements of the singularity as labeled in Fig. 1a. MgO (100) has a cubic unit cell with a fourfold rotational symmetry and therefore shows a modulation with a periodicity of $\pi/2$. For $\delta = 0$, there is no difference in absorption of left- and right-handed helical light and the curves overlap (Fig. 1b). When the singularity is shifted to either side from the center of the beam, the dominance of absorption switches between the two helicities of light (Fig. 1c, d). The orientation-dependent transmission is invariant of the position of the singularity. However, the magnitude of HD (Type I) signal remains unaffected with crystal orientation but depends on the position of the singularity. Similar results were obtained for ZnO (11–20) and α-quartz (z-cut) (Supplementary Section 2). Orientation dependence in crystals can also be obtained with non-OAM beams[29] (Supplementary Section 2) and via high harmonic generation[3,4] in nonlinear regime. Extraction of crystal symmetry is due to the multiphoton nature of the interaction and direction dependence of the effective mass of the electron that acts as a local probe[29].

### Amorphous solids

In a-solids, dichroism and modulation in nonlinear absorption are not expected due to the absence of long-range order. However, Fig. 2a

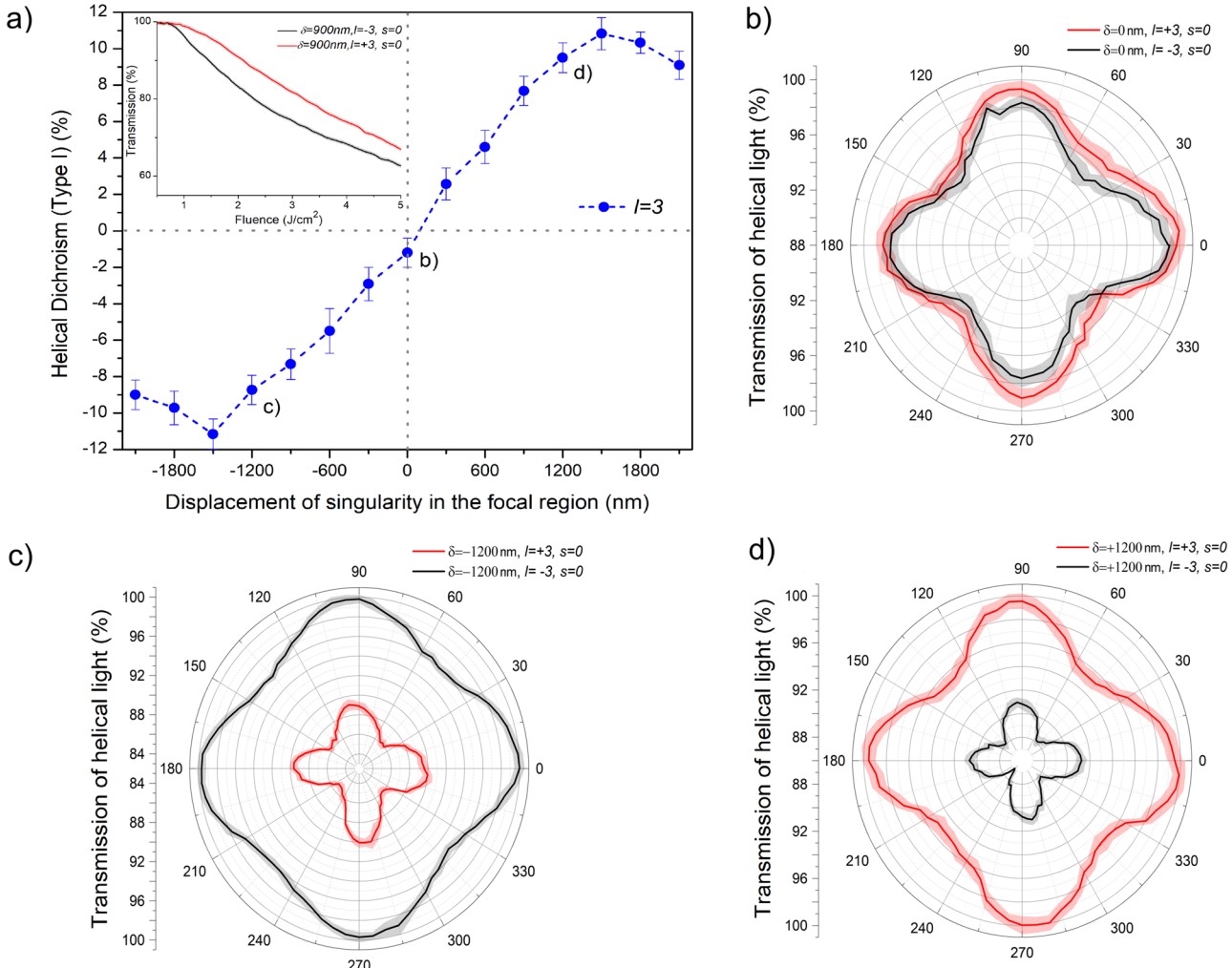

**Fig. 1 | Helical dichroism and orientational dependent transmission of helical light in a crystal. a** HD(Type I) = A( + l, s) − A( − l, s) for crystalline MgO (100) as a function of the displacement of the singularity (δ) in linearly polarized (ε = 0.05) asymmetrical OAM beam (l = ± 3). Inset shows transmission of l = ± 3 beam as a function of peak laser fluence for δ = 900 nm. **b–d** Orientation-dependant transmission of l = ± 3 in MgO at the position of displaced singularities marked in (**a**). The error bars in (**a**) represent the standard error, of multiple measurements (n = 3), calculated for an average fluence range used to obtain HD (see text for details). The error bands in (**b–d**) represent the standard error at every position of the crystal for n = 20 points.

shows that dichroism exits even in a-solids. For asymmetric helical beams, HD (Type I) in fused silica and borosilicate glass, plotted as a function of the displacement of the singularity, shows similar behavior as in c-solids (Fig. 1a). Figure 2b shows orientation-dependent transmission of helical light in fused silica (for δ = − 1200 nm), confirming the lack of symmetry since periodic modulation is absent at all positions of the singularity. However, nonlinear absorption of light with opposite helicities is different representing the HD (Type I) shown in Fig. 2a. The results of Figs. 1 and 2 suggest that helical dichroism is independent of the crystal orientation and is intrinsic to both materials. This is in contrast to conventional understanding that dichroism with circularly polarized light is absent in disordered solids (Supplementary Section 5).

Helical dichroism in liquids was recently shown to arise from coupling between the electric dipole and the electric quadrupole transition terms[25,30]. In disordered solids, this coupling term is expected to vanish upon isotropic averaging while it can survive in gas and liquid phases due to laser-induced dipole force. The presence of HD (Type I) in a-solids and its resemblance to c-solids, while displaying no symmetry in the orientation-dependent transmission due to the absence of long-range order, suggests that HD (Type I) can be attributed to short- to medium-range order. The origin of HD in solids can be

understood in terms of electron transitions via Multiphoton-assisted tunneling (MPAT). The electron displacement during the MPAT serves as a local probe of the short- and medium-range order whereas the helicity dependence arises from higher-order multipoles.

In MPAT process, nonlinear absorption of the incident light is first initiated by multiphoton absorption which promotes an electron from the valence band to an intermediate excited state within the bandgap from which tunneling to conduction band takes place. In a-solids, such intermediate states are the so-called band tail states that extend into the bandgap. Long-range disorder leads to localized states near the bottom (top) of the conduction (valence) band and their density decreases exponentially away from the band edges. In addition, defects, impurities and dangling bonds give rise to localized states in the middle of the bandgap[13].

MPAT occurs at moderate intensities where the nonlinear absorption is not dominated by either ground-state tunneling or multiphoton transition alone. These two regimes are typically identified by Keldysh parameter[31] given by Eq. (3) (see "Electron displacement during interband transition" for details). Multiphoton (or tunneling) transition is dominant when the Keldysh parameter is larger than ~2 (or smaller than 1)[32–34]. In fused silica, the Keyldysh parameter corresponding to our experimental intensities used to

obtain HD are in the range of -1.1–2.5 where MPAT process is predominant resulting in strong-field induced interband excitation.

In ground-state tunneling, the electron that is promoted from valence to conduction band gains a spatial displacement ($\mathbf{x}_0$) that is proportional to the bandgap $E_g$ and can be approximated by Eq. (2) ("Electron displacement during interband transition"). In MPAT process, $E_g$ has to be replaced by the energy difference between the band tail state and conduction band edge. This leads to a smaller $\mathbf{x}_0$ compared to the upper limit set by tunneling from valence to conduction band, which is in the range of 6–18 Å for fused silica ($E_g \approx 9 ev$[34,35]). In comparison, the short- to medium-range order in fused silica is <20 Å[9,14–16]. The total electron displacement which is a sum of the displacements during tunneling from the intermediate state to the conduction band and multiphoton absorption from valence to intermediate state, is within this range.

Therefore, in a-solids photons are absorbed in a quasi-ordered environment over distances defined by the short- to medium-range order. This results in nonzero isotropic averaging of molecular response tensor. The three-dimensional network of $SiO_2$ could still consist of periodic clusters each oriented in a different direction. However, these clusters of ordered molecules will absorb the incident light at varying degrees depending on the overlap between the laser polarization and their absorption transition moments[36]. Therefore, the total absorbed energy by these clusters give rise to a finite contribution towards interband transitions and a nonzero $l$-dependence (see "Origin of Helical dichroism in solids" for details). As a result, the behavior of HD (Type I) signal is the same in disordered and ordered solids. In c-solids, the ordered environment always ensures nonzero isotropic averaging where HD(Type I) arises from electric dipole electric quadrupole coupling term. Electron transitions via MPAT process can still play a role where the intermediate states could be due to degenerate exciton states, defects, impurities, and boundary effects[37–39].

### Chiral crystalline solids

C-solids also exhibit chirality, typically studied using sophisticated diffraction and imaging techniques[6,7]. Conventional CD using Gaussian light (Supplementary Section 5) is inefficient as it is convoluted by competing signals from optical and material properties[5]. Using helical light we show in Fig. 3 that the handedness of a chiral solid can be probed. Figure 3a shows HD (Type I) signal in crystalline (z-cut) left-quartz (L) and right-quartz (R) for different positions of the singularity in OAM beam. The spatial variation of HD(Type I) is similar to other crystals and amorphous solids (Figs. 1a and 2a). A key difference is that with asymmetric OAM beam, the magnitude of helicity-dependent absorption in the two chiral structures is different. This difference suggests that chiral solids exhibit another type of dichroism, defined as HD (Type II: $\pm l; \pm s) = 2 \frac{R(\pm l; \pm s) - L(\pm l; \pm s)}{R(\pm l; \pm s) + L(\pm l; \pm s)}$. HD (Type II) is the difference in absorption of a specific-handed helical light in the left- and right-handed chiral solid. It represents a chiral light–chiral matter interaction and is odd under PT symmetry.

HD (Type II) signal, shown in Fig. 3b, as a function of peak laser fluence changes sign with the helicity of the incident light enabling to efficient differentiation the handedness of crystal structure. This chiral signal, obtained at a specific position of the singularity marked by the rectangle in Fig. 3a, is more prominent with asymmetric OAM beam and is weakly influenced by the position of the singularity within the OAM beam. Chiral signal using circularly polarized Gaussian beam (with $l = 0$, $s = \pm 1$) is also shown in Fig. 3b and demonstrates the signal fluctuations with lower average efficiency. The obtained chiral signal is an order of magnitude higher than the reported values from solid-state-based CD in complex chiral crystals[40,41] and is comparable to recently demonstrated hard X-ray-based HD in powdered molecular media[24].

Another representation of HD (Type II) is the orientation-dependent transmission in different-handed quartz for specific helicity shown in Fig. 3c, obtained when the singularity is at the positions marked by a solid rectangle in Fig. 3a. A modulation of $\pi/3$ shows the sixfold rotational symmetry for quartz (Supplementary Section 2). This shows that transmission for specific helicity and position of singularity is different for both chiral solids. Therefore, crystal structure and chirality can be probed simultaneously.

### Bandgap dependence of helical dichroism

Figure 4 demonstrates HD (Type I) in materials with different bandgaps. Figure 4a, shows spatial variation of HD (Type I) as a function of displacement of the singularity, for $TeO_2$ (red, bandgap of 3 eV)[42], ZnO (cyan, 3.4 eV)[43], MgO (black, 7.7 eV)[44], fused silica (magenta, $9 ev$[34,35], and quartz (blue, 9.5 eV)[45]. The different magnitudes of HD(Type I) suggest it is a material-dependent property and HD signal increases with the bandgap. This material dependence can also be observed in

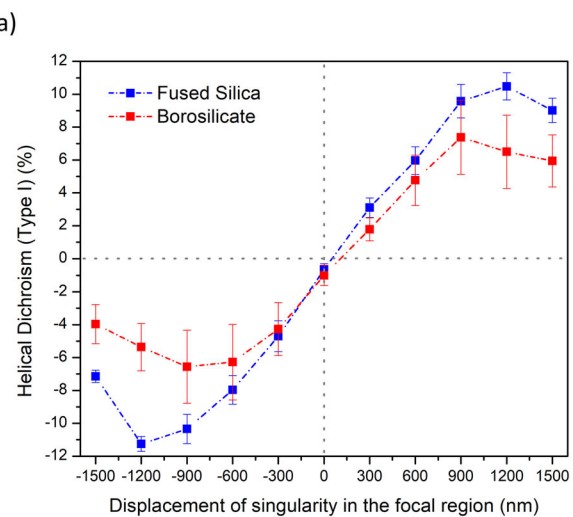
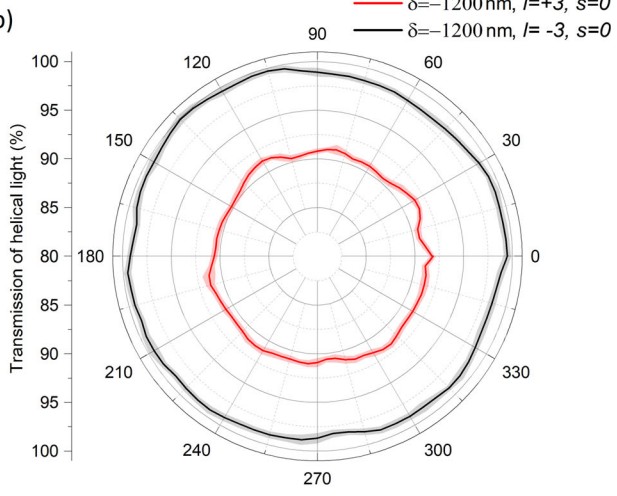

**Fig. 2 | Helical dichroism in amorphous solids. a** HD (Type I) for fused silica (blue) and borosilicate (red) as a function of displacement of the singularity with linearly polarized ($\epsilon = 0.05$) asymmetrical OAM beam ($l = \pm 3$). **b** Orientation-dependent transmission of $l = \pm 3$ in fused silica at $\delta = -1200$ nm. The error bars in (**a**) represent the standard error, of multiple measurements ($n = 3$), calculated for an average fluence range used to obtain HD. The error bands in (**b**) represent the standard error at every position of the crystal for $n = 20$ points.

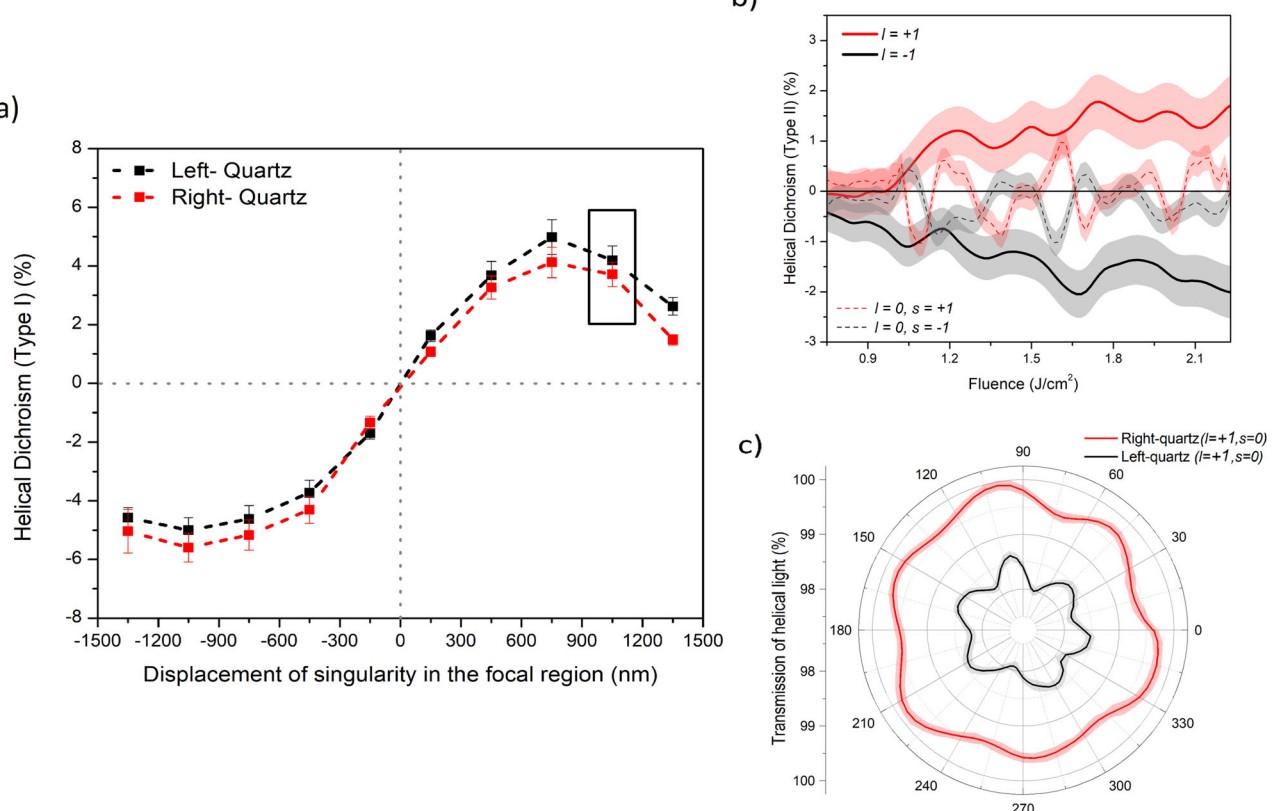

**Fig. 3 | Probing chirality in quartz with linearly polarized ($\epsilon = 0.05$) asymmetrical OAM beams. a** HD (Type I) for left- and right-handed quartz as a function of displacement of the singularity ($l = \pm 1$). **b** Chiral signal, HD (Type II: $\pm l; \pm s$) = 2 $\frac{R(\pm l; \pm s) - L(\pm l; \pm s)}{R(\pm l; \pm s) + L(\pm l; \pm s)}$ as function of peak laser fluence for linearly polarized $l = \pm 1$ and circularly polarized $l = 0$ beams. **c** Orientation-dependent transmission of $l = +1$ in right- and left-handed in quartz. **b**, **c** were obtained when the singularity was at the position marked by a solid rectangle in (**a**). The error bars in (**a**) represent the standard error, of multiple measurements ($n = 3$), calculated for an average fluence range used to obtain HD. The error bands in (**b**) represent the error propagation of the standard error of multiple transmission curves. The error bands in (**c**) represent the standard error at every position of the crystal ($n = 20$ points).

our qualitative model discussed in "Origin of Helical dichroism in solids". Figure 4b, depicts an average HD (Type I) signal, for three $+\delta$ positions (600, 900, and 1200 nm) of Fig. 4a, plotted as a function of bandgap.

**Control and tunability of HD (Type I)**

Figure 5 shows simulated and experimental curves obtained by (i) superimposing Gaussian and OAM beams, and (ii) varying the $l$-value. To model HD (Type I), we considered the total rate of interband transition ($W_{cv}^{\pm}$) evaluated using Eq. (20). It is a product of multiphoton transition probability amplitude from the ground to the intermediate state and tunnel ionization probability amplitude from the intermediate state to the conduction band. HD (Type I) is defined in terms of differential averaged energy absorbed $\Delta\Gamma$, normalized with respect to the incident laser energy (see "Origin of Helical dichroism in solids" for details).

$$\Delta\boldsymbol{\Gamma} = \Gamma^{+} - \Gamma^{-} = \mathcal{D}(\Upsilon^{+} - \Upsilon^{-}) = \mathcal{D}\left( \mathrm{Re}\left[\nabla_i E_j^{+} E_i^{+*}\right] - \mathrm{Re}\left[\nabla_i E_j^{-} E_i^{-*}\right] \right) \quad (1)$$

where $\Upsilon^{\pm}$ is optical helicity which describes the handedness of helical light and $E_{i,j}$ is the electric field described by Eq. (23) ("Superposition of OAM and Gaussian" and Supplementary Section 7 for details). $\mathcal{D}$ represents a collection of physical quantities (given by Eqs. (18)–(20)) that are independent of helicity within our approach, and can be estimated to be $7.3 \times 10^{-5}$ ("Origin of Helical dichroism in solids" and Supplementary Section 9). Therefore, HD (Type I) is a beam-dominated property and does not exist for symmetric LG beams

($\delta = 0$). For asymmetric LG beams ($\delta \neq 0$), the electric–magnetic dipole coupling term in Eq. (17) vanishes (because we take the difference between the left- and right-helical beams for the same polarization), and the electric dipole–quadrupole coupling term is nonzero resulting in HD (Type I). The electric dipole–quadrupole coupling term also contains the gradient of the electric field giving rise to $l$-dependence, as can be seen in Fig. 5c. The simulation curves shown in Fig. 5 are plotted by integrating HD (Type I) over the whole beam cross-section, HD (Type I) = $\int_{-w_0}^{w_0} \Delta\boldsymbol{\Gamma}\,\mathrm{d}x\mathrm{d}y$.

For simplicity, we treated intermediate state tunneling within the dipole approximation and reduced the multiphoton absorption to a single-photon transition expanded to higher-order multipoles. We simulate HD(Type I) for different ratios of superposition of Gaussian and OAM beams (Fig. 5a) and $l$-values (Fig. 5c) using the above equation. For a specific $l$-value of an asymmetric OAM beam, the magnitude of HD (Type I) signal is maximum for a pure OAM beam and decreases as the amount of Gaussian beam in the superposition increases (Fig. 5a). In case of linearly polarized light, HD signal vanishes for a pure Gaussian beam. Experimental results shown in Fig. 5b for fused silica agrees with the simulations (Fig. 5a). Also, simulated and experimental HD signals scale with $l$-value as shown in Fig. 5c, d for two different $l$-values. Similar control and tunability of HD was also observed in c-solids such as MgO (Supplementary Section 3). The simulated HD values differ by few orders of magnitude compared to experimental values. This variation could be due the approximate values used for electric dipole and quadrupole moment, and material response.

a)

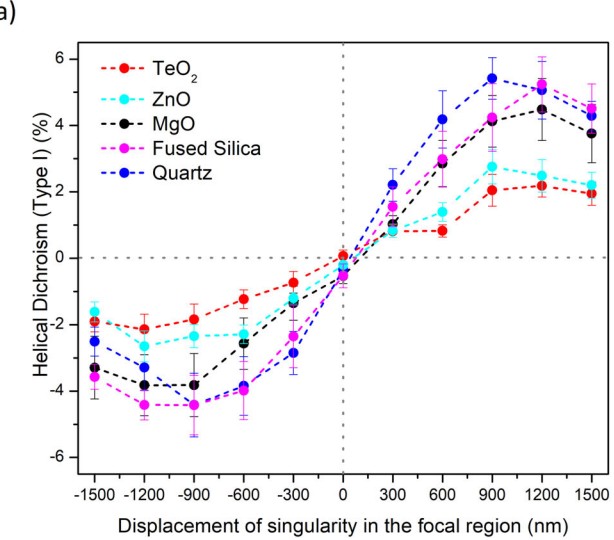

b)

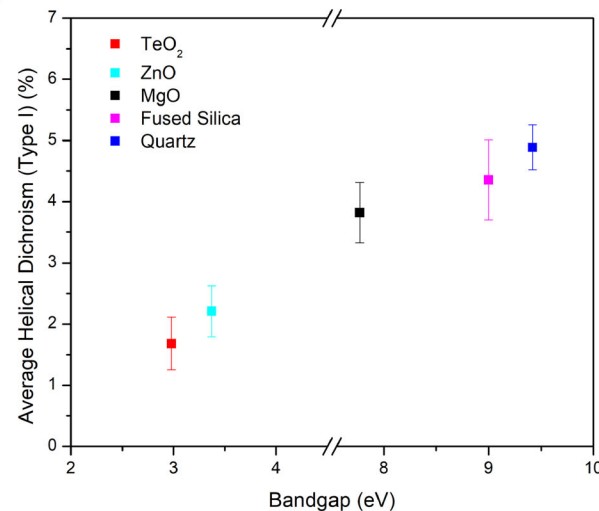

**Fig. 4 | Bandgap dependence of helical dichroism. a** HD(Type I) in different materials as a function of displacement of singularity in the OAM beam for $l = \pm 1$ in crystalline and amorphous solids. **b** Average magnitude of HD (Type I) as a function of bandgap for $l = \pm 1$. The error bars in (**a**) represent the standard error, of multiple measurements ($n = 3$), calculated for an average energy range used to obtain HD. The error bars in (**b**) represent the average of three $\delta$ positions in (**a**) where the HD (Type I) signal is maximum. All curves are obtained under identical experimental conditions.

HD signal changes in discrete steps when $l$-value is varied but can be tuned almost continuously by changing the superposition of OAM and Gaussian beams. In addition, HD signal can also be varied by changing the ellipticity of laser polarization[25]. These are the features of our technique that has no comparable equivalent in other existing solid-state-based chiroptical techniques. Continuous tunability of HD by superposition of OAM and Gaussian beams by electrical detuning of optical retardation offers instrumental advantage. A single higher-order q-plate is sufficient to reproduce the results of all lower-order q-plates by controlling the voltage on the q-plate.

## Discussion

In conclusion, we demonstrate HD in both amorphous and crystalline solids including chiral systems. HD in the condensed phase provides valuable information on chirality. Conventional solid-state chiroptical techniques such as CD have poor efficiency because the chiral signal is weaker than the artifact signals from macroscopic anisotropies such as linear birefringence. The efficiency of HD enables to achieve chiral recognition on a solid surface and is of fundamental importance in various fields of surface and material sciences. In addition, the control and tunability of HD provides new opportunities in the development of enantioselective catalysis[46] and asymmetric synthesis of bioactive molecules[47], chiral sensors[48,49], and molecular electronic devices[50,51]. In laser processing of transparent materials, localized changes induced in the bulk can result in structural changes on sub-micron dimensions confined to the laser focal volume[52,53]. These changes modify the bandgap and are hard to study due to lack of in situ probes. Pump-probe spectroscopy involving Gaussian and OAM beams can shed light on such changes by monitoring the magnitude of HD signal due to its dependence on the bandgap.

The presence of intrinsic dichroism in amorphous materials is due to the existence of short- to medium-range order. Material response to the phase of light associated with OAM beams will aid the efforts to understand the mysterious nature of amorphous materials. Furthermore, HD can be extended to conjugated polymers that are used as active materials in devices for printed and flexible organic electronics[54]. Transport properties in such polymers is widely believed to be due to long-range order, so research focused primarily on increasing the crystallinity of polymers. However, recent studies showed that local aggregation over few chains is sufficient to ensure high mobility[55]. In other words, the key to designing high-mobility polymers is not to increase their crystallinity but rather to improve their tolerance for disorder. Typically, information on short-range ordering in polymers is often achieved by means of radial distribution function derived from X-ray diffraction[56]. HD can shed light on short-range order that influences electronic properties in polymers and semiconducting alloys.

## Methods

### Differential absorption measurements

Transmission measurements were performed using a Ti: Sapphire laser amplifier system, operating in an external trigger mode producing 45 fs, 800 nm pulses with a maximum pulse energy of 2.5 mJ. An aspheric objective lens (NA = 0.3) was used to focus the femtosecond pulses into solid samples with typical dimensions of $10 \times 10 \times 1$ mm. A second aspheric objective with the same or higher numerical aperture (NA = 0.5) collected and collimated the transmitted light onto a photodiode (PD2), positioned immediately after the objective (Supplementary Section 1 for a schematic of the experimental setup). For every laser shot, the transmitted light signal on PD2 was normalized with the incoming light signal on PD1, reflected off a glass plate positioned in the beam path at an angle of ~20° to avoid Brewster's angle. The signals generated by PD1 and PD2 were stretched by an electronic pulse stretcher, discretized, and recorded by a data acquisition card. A combination of a half-wave plate and a polarizer was used to vary the pulse energy (Supplementary Section 1). The incident pulse energies were measured before the objective. During the measurement, for every laser shot, the pulse energy was increased by ~3 nJ and the sample was translated by 5 μm to irradiate a fresh sample. Multiple transmission curves similar to the inset of Fig. 1a were obtained for each sample to be averaged and smoothed. The difference in the normalized transmission of left- and right-helical light is proportional to the differential absorption. To ensure shortest pulses in the interaction region, a negative chirp was introduced and optimized by measuring the second harmonic generation in a BBO crystal placed at the location of the sample. A single-shot auto-correlator then continuously monitored the pulse duration. The pulse duration at the interaction region is about 100 fs. Prior to each experiment,

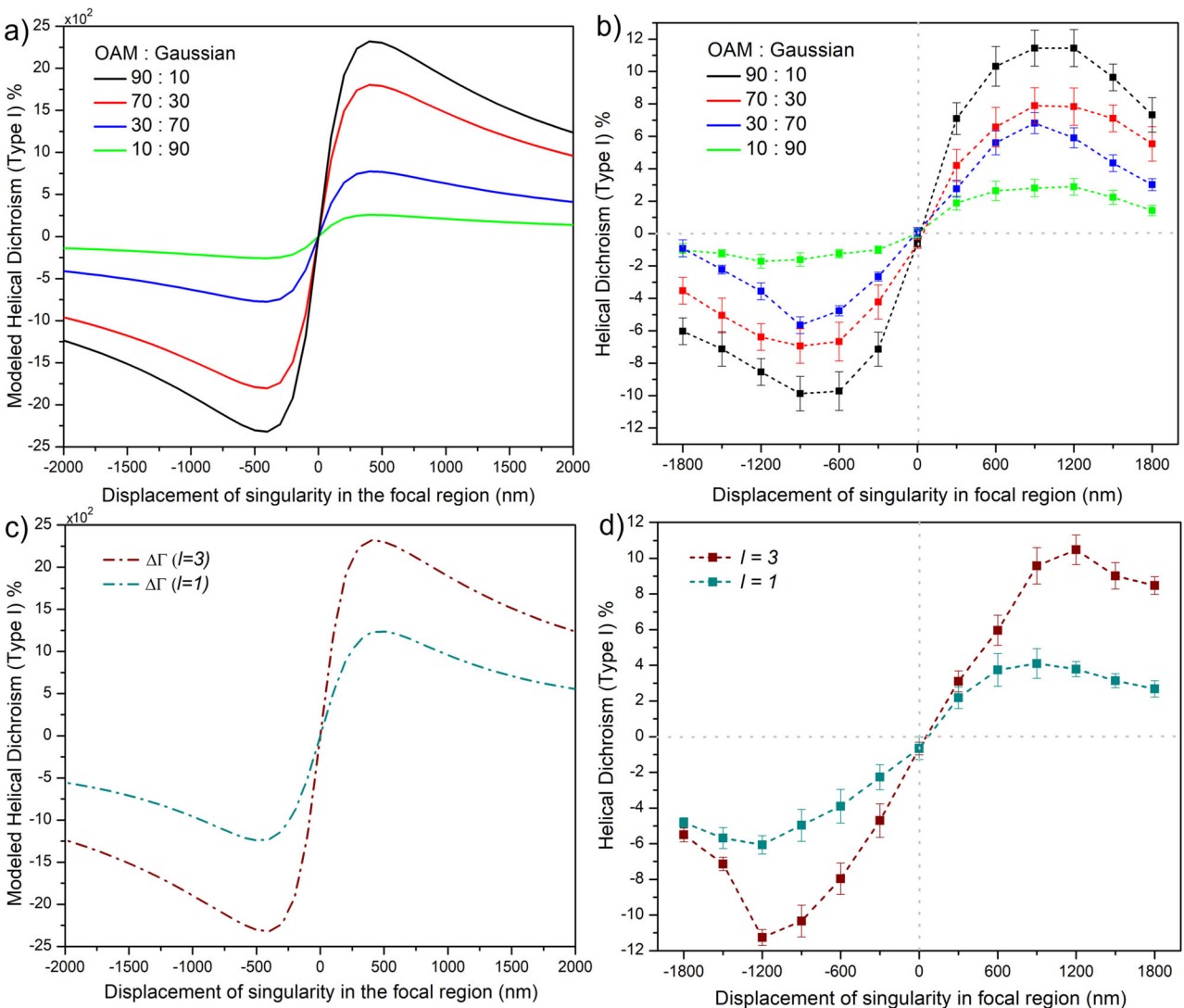

**Fig. 5 | Tunability of Helical dichroism.** Simulated and measured HD (Type I), in fused silica as a function of the displacement of singularity for (**a**, **b**) variable ratios of superposition of linearly polarized ($\epsilon = 0.05$) OAM ($l = \pm 3$) and Gaussian beams, and **c**, **d** linearly polarized ($\epsilon = 0.05$) helical light with $l = \pm 1, \pm 3$. In simulation, displacement of the singularity was considered perpendicular to the polarization following experimental conditions. The error bars in (**b**, **d**) represent the standard error, of multiple measurements ($n = 3$), calculated for an average fluence range used to obtain HD. The HD (Type I) was modeled using the relation $\int_{-w_0}^{w_0} \Delta \boldsymbol{\Gamma}\, dx, dy = \int_{-w_0}^{w_0} \mathcal{D}(\Upsilon^+ - \Upsilon^-)\, dx, dy$ where $\mathcal{D}$ represents a collection of physical quantities and $\Upsilon^{\pm}$ is optical helicity which describes the handedness of helical light (see text and "Methods" for details).

transmission measurements were always performed without a sample (in air) to determine and minimize background errors resulting from any discrepancies between the photodiodes (Supplementary Section 6). In addition, the measured single-shot beam profile, pulse spectrum, and OAM value remained unchanged after transmission through the samples[25]. The crystal samples were procured from MTI Corp., MSE Supplies and chiral quartz from Knight Optical. Fused silica was bought from SPI Supplies.

### Generation of OAM beams
Light beams carrying orbital and/or spin angular momentum were generated and controlled by OAM/SAM unit (Supplementary Fig. 1) consisting of a combination of half- and quarter-wave plates (HWP and QWP), linear polarizer (LP) and a birefringent liquid crystal based phase plate called $q$-plate[21,28,57]. When an incident Gaussian beam propagates through the $q$-plate with a topological charge $q$, it acquires an OAM defined by $l = \pm 2q$ with a phase singularity and hence a null intensity region at the center of the beam - an optical vortex. The

wavefront structure of such beams undergo $l$ intertwined rotations in one wavelength, the direction of rotation is determined by the sign of the input polarization. The conversion efficiency of the $q$-plates were $91 \pm 2\%$ for $l = 1, 3$. Linearly polarized OAM light $s = 0, l = \pm 1$ was generated using a combination of QWP, $q$-plate, QWP, and LP. The ellipticity of linearly polarized OAM light was $5 \pm 2\%$ reaching the sample.

### Displacement of singularity
The singularity/null intensity region in the OAM beam was displaced by translating the $q$-plate, mounted on a x, y-stages with a step size of $250 \pm 10\,\mu m$. When focused by the objective, this translated to a displacement step size of $300 \pm 20$ nm with respect to the center of the beam. The calibration was achieved by measuring the total translation required to displace the singularity to the periphery of the defocused beam and comparing it to the measured spot size of $2 \pm 0.2\,\mu m$ obtained by knife-edge measurements. The helical vortex beams can be further classified into vector and scalar vortex beams. The superposition of vector vortex beams with linear Gaussian leads to the

shifting of singularity towards the periphery of the beam for $l = 1$ and splitting of the singularity for higher-order $l$-value[58]. The superposition of linear Gaussian with scalar vortex beam causes the null intensity region to fade and does not lead to the shifting of singularity, which was used in the experiments.

## Electron displacement during interband transition

Classically, in atoms, tunneling results in spatial displacement of the electron given by $\mathbf{x}_0 = I_p/(e\mathbf{E})$, where $I_p$ is the ionization potential[31,59]. $\mathbf{x}_0$ is the distance traversed by the electron through the barrier to the tunnel exit. This concept can also be extended to solids. The energy absorbed from the incident laser fields is proportional to the associated injection current given by $\mathbf{J} = en_0\mathbf{x}_0\dot{\rho}$[60], where $e$ is the electron charge, $n_o$ is electron density in the conduction band, $\dot{\rho}$ is the injection rate which is proportional to the transition rate $W$, $\mathbf{x}_0$ is the spatial displacement of the electrons after being promoted from the valence band to the conduction band. (**J**) is fundamental to maintain energy conservation ($\dot{u} = -\mathbf{J}\cdot\mathbf{E}$). In optical tunneling, energy conservation is fulfilled when this current reflects a spatial displacement of the electrons given by[60]

$$\mathbf{x}_0 = \mathbf{E}E_g/\left(eE^2\right) \quad (2)$$

where $E_g$ is the bandgap and $\mathbf{E}$ is the electric field. We assume a parabolic conduction band to eliminate contributions from band anharmonicities and Bloch oscillations due to reflections at zone boundaries[60].

Injection current was recently used to explain the origin of strong-field-induced harmonic generation in fused silica[60]. In amorphous fused silica ($E_g \approx 9$ ev[34,35]), $\mathbf{x}_0$ varies from 6 to 18 Å for the laser intensities used in our experiments. Transition of electron from valence to conduction band can occur via two mechanisms, multiphoton or tunneling, depending on the laser intensity. They are typically identified by the Keldysh parameter ($\gamma$)[31,33]. For solids, it is defined as

$$\gamma = \frac{\omega}{e}\left[\frac{mcn\varepsilon_0 E_g}{I}\right]^{1/2} \quad (3)$$

where $\omega$ is the laser frequency, $I$ is the incident laser intensity (Supplementary Section 4), $m$ and $e$ are the reduced mass and charge of the electron, $c$ is the velocity of light, $n$ is the refractive index of the material, $E_g$ is the bandgap of the material and $\epsilon_0$ is the permittivity of free space. When $\gamma > 2$, interband excitations are dominated by multiphoton process and when $\gamma < 1$ it is dominated by tunneling.

For the laser intensities in the range of $2 \times 10^{13}$ to $8 \times 10^{13}$ W/cm$^2$ that were used in the experiments, Keldysh parameter varied from ~2.5 to 1.1 for fused silica. In the intermediate regime when $1 \lesssim \gamma \lesssim 2$ the electron transitions are dominated by multiphoton-assisted tunneling (MPAT)[33,61]. The MPAT process, developed for atoms, can be extended to solids and visualized as electron transition from valence band to an intermediate state by mulitphoton absorption and subsequent tunneling from that state to the conduction band[31,61].

In MPAT process, there will be two contributions to electron displacement—one from multiphoton transition to the intermediate state and the other from subsequent tunneling to the conduction band ($\mathbf{x}_0$ is given by Eq. (2) where $E_g$ needs to be replaced by the energy gap between intermediate and conduction band states). However, $\mathbf{x}_0$ in MPAT will be smaller compared to direct tunneling transitions from valence band to conduction band which sets the upper limit for electron displacement[32,61]. $\mathbf{x}_0$ in MPAT is comparable to short-medium-range order in fused silica which is < 20 Å[9,14–16]. As a result, isotropic averaging is restricted to an ordered environment giving rise to finite contribution in a-solids. In c-solids, the ordered environment always ensures a nonzero HD signal upon isotropic averaging. Moreover, our transmission curves reproduce the crystal symmetry due to its

dependence upon the injection current, similar to the results that were obtained from the high harmonic yields[60].

## Origin of Helical dichroism in solids

Helical Dichroism in solids can be understood in terms of molecular multipole moments during light-matter interaction. When the incident laser intensity is not sufficient for direct interband tunneling, electron transitions are facilitated by multiphoton excitation to intermediate states and subsequent tunneling to the conduction band. The transition amplitude for MPAT process involving a single-photon absorption (SPA) from the valence band ($\psi_v$) to an intermediate excited state ($\psi_m$) and subsequent tunneling to the conduction band ($\psi_c$) is given by (Supplementary Section 8 for further details)

$$M_{cv} = \sum_m \underbrace{\frac{\langle\psi_m|\hat{V}_i|\psi_v\rangle}{\hbar(\omega_{mv}-\omega)}}_{M_{SPA}}\underbrace{\frac{i}{\hbar}\int_{t_0}^t d\tau\langle\psi_c(\tau)|\hat{V}_f(\tau)|\psi_m(\tau)\rangle e^{i(\omega_{mv}-\omega)\tau}}_{M_{TNL}} \quad (4)$$

where $\omega$ is the laser frequency, $\omega_{mv} = (E_m - E_v)/\hbar$, $E_m$, and $E_v$ are the energies of the intermediate and valence band states, respectively. In addition, $\hat{V}_i$ and $\hat{V}_f$ are multiphoton and tunneling interaction Hamiltonians written in multipole expansion and dipole approximation, respectively. The MPAT transition amplitude is a product of two factors; a time-independent single-photon transition amplitude $M_{SPA}$ and a time-dependent excited state tunneling amplitude $M_{TNL}$. The interaction Hamiltonian is given by $\hat{V}(t') = \frac{1}{2m}[2e\mathbf{p}\cdot\mathbf{A}(R,t) + e^2\mathbf{A}^2(R,t)]$, where the $m$ is the electron mass, $\mathbf{p}$ is the momentum vector in the molecular frame ($\mathbf{r}$) and $\mathbf{A}(R,t)$ is the vector potential in the laboratory frame ($\mathbf{R}$).

In crystals, the eigenstates are written in terms of bloch wavefunctions (plane wave modulated by the crystal periodicity function $u_k(\mathbf{r})$. For a-solids, we can approximate the eigenstates as bloch-like wavefunctions (as long as the electron displacement is within $\mathbf{x}_0$[13]) with effective wave vectors $\mathbf{k}'$, $\mathbf{k}''$ and $\mathbf{k}$ for valence band state, intermediate state and the conduction band state, respectively. The valence band and intermediate state wavefunctions are given by

$$|\psi_v(\mathbf{r},\tau)\rangle = u^v_{\mathbf{k}'}(\mathbf{r})e^{i\mathbf{k}'\cdot\mathbf{r}}e^{-i\frac{E_v}{\hbar}\tau} \quad (5)$$

$$|\psi_m(\mathbf{r},\tau)\rangle = u^m_{\mathbf{k}''}(\mathbf{r})e^{i\mathbf{k}''\cdot\mathbf{r}}e^{-i\frac{E_m}{\hbar}\tau} \quad (6)$$

The conduction band state is approximated by a Volkov-type solution[62]

$$|\psi_c(\mathbf{r},\tau)\rangle = u^c_{\mathbf{k}}(\mathbf{r})\exp\left[i\left(\mathbf{k}\cdot\mathbf{r} - \frac{1}{\hbar}\int_0^\tau E_c[\mathbf{k}(t'')]dt''\right)\right] \quad (7)$$

$$E_c[\mathbf{k}(t'')] = E_g + \frac{1}{2m_1}(\hbar\mathbf{k} - e\mathbf{A}(t''))^2 \quad (8)$$

where $E_v = \frac{\hbar^2 k^2}{2m_0}$, $m_0$ and $m_1$ are the effective mass of the valence and the conduction bands, $E_g$ is the bandgap, $\mathbf{r}$ is the spatial coordinate in molecular frame and $\mathbf{A}$ is the vector potential in laboratory frame given by ($\mathbf{A}(t) = A_0(\alpha\cos\omega t\hat{x} + \beta\sin\omega t\hat{y})$) for elliptical polarization. In our intensity range, the above Volkov-like wavefunction can be limited to a linear dependency of the vector potential. Solving the time integral leads to

$$|\psi_c(\mathbf{r},\tau)\rangle = \frac{u^c_{\mathbf{k}}(\mathbf{r})}{(2\pi)^{3/2}}e^{i\mathbf{k}\cdot\mathbf{r}-\frac{i}{\hbar}(E_g\tau+E_c\tau+\gamma_0\hbar[k_x\alpha\sin(\omega\tau)-k_y\beta\cos(\omega\tau)])} \quad (9)$$

where $E_c = \frac{\hbar^2 k^2}{2m_1}$ is the kinetic energy of the conduction band and $\gamma_0 = \frac{eA_0}{m_1\omega}$ is the quiver amplitude. Substituting all the above states (Eqs. (5)–(7)) in the transition amplitude (Eq. (4)) and taking the

evolution of states over $t_0 = -T/2$, $t = T/2$, with the limit $T \to \infty$[62,63], we obtain

$$M_{cv} = \frac{iA_0 e}{mh(2\pi)^{3/2}} \sum_m \frac{1}{h(\omega_{mv} - \omega)} \int_{-\infty}^{\infty} u_{\mathbf{k}''}^{m*}(\mathbf{r}') \hat{V}_i u_{\mathbf{k}'}^v(\mathbf{r}') e^{i(\mathbf{k}' - \mathbf{k}'') \cdot \mathbf{r}'} d^3 r' \int_{-\infty}^{\infty} u_{\mathbf{k}}^{c*}(\mathbf{r}) u_{\mathbf{k}''}^m(\mathbf{r}) e^{i(\mathbf{k}'' - \mathbf{k}) \cdot \mathbf{r}} d^3 r$$
$$\times \lim_{T \to \infty} \int_{-T/2}^{T/2} e^{\frac{i}{\hbar}(E_g + \epsilon - \hbar\omega)\tau} (\alpha p_x \cos\omega\tau + \beta p_y \sin\omega\tau) e^{i\gamma_0 [k_x \alpha \sin(\omega\tau) - k_y \beta \cos(\omega\tau)]} d\tau$$

(10)

Solving the time integral using Jacobi–Anger expansion $\cos\omega t e^{i\alpha\sin\omega t} = \frac{1}{\alpha} \sum_{n=-\infty}^{\infty} n J_n(\alpha) e^{in\omega t}$, we obtain

$$M_{cv} = \frac{iA_0 e}{mh(2\pi)^{3/2}} \frac{1}{\gamma_0} \sum_m p_{mv}^{SPA} \sum_{\eta, \zeta = -\infty}^{\infty} i^\zeta \int_{-\infty}^{\infty} u_{\mathbf{k}}^{c*}(\mathbf{r}) \left[ \frac{p_x}{k_x} \eta + \frac{p_y}{k_y} \zeta \right] u_{\mathbf{k}''}^m(\mathbf{r}) e^{i(\mathbf{k}'' - \mathbf{k}) \cdot \mathbf{r}} d^3 r$$
$$\times J_\eta(-\gamma_0 k_x \alpha) J_\zeta(\gamma_0 k_y \beta) \left[ \lim_{T \to \infty} \frac{2\hbar \sin\left( (E_g + \epsilon - N\hbar\omega) \frac{T}{2\hbar} \right)}{(E_g + \epsilon - N\hbar\omega)} \right]$$

(11)

where $\epsilon = \frac{\hbar^2 k^2}{2m^*}$ is the total kinetic energy, $m^{*-1} = m_1^{-1} - m_0^{-1}$ is the total effective electron mass, and $N\hbar\omega = (\eta + \zeta)\hbar\omega + \hbar\omega$ is the total number of photons absorbed during the MPAT process. Use of Jacobi–Anger expansion allowed us to discretize the tunneling contribution and, for analytical analysis, treated the transition from intermediate state to conduction band state non-perturbatively. $p_{mv}^{SPA}$ represents the single-photon absorption probability amplitude given by

$$p_{mv}^{SPA} = \frac{1}{\hbar(\omega_{mv} - \omega)} \int_{-\infty}^{\infty} u_{\mathbf{k}''}^{m*}(\mathbf{r}') \hat{V}_i u_{\mathbf{k}'}^v(\mathbf{r}') e^{i(\mathbf{k}' - \mathbf{k}'') \cdot \mathbf{r}'} d^3 r'$$

(12)

We further define the tunneling probability amplitude as

$$p_{cm}^{TNL} = \frac{m_1}{mh(2\pi)^{3/2}} \sum_{\eta, \zeta = -\infty}^{\infty} i^\zeta J_\eta(-\gamma_0 k_x \alpha) J_\zeta(\gamma_0 k_y \beta) \int_{-\infty}^{\infty} u_{\mathbf{k}}^{c*}(\mathbf{r}) \left[ \frac{p_x}{k_x} \eta + \frac{p_y}{k_y} \zeta \right] u_{\mathbf{k}''}^m(\mathbf{r}) e^{i(\mathbf{k}'' - \mathbf{k}) \cdot \mathbf{r}} d^3 r$$

(13)

Using the property $\lim_{T \to \infty} T^2 \frac{\sin^2(x)}{x^2} = 2\pi\hbar T \delta(\epsilon + E_g - N\hbar\omega)$ where $x = (E_g + \epsilon - N\hbar\omega)\frac{T}{2\hbar}$[64], the transition rate from the valence to the conduction band, defined as $W_{cv} = \frac{d}{dt}|M_{cv}|^2$, can be expressed as

$$W_{cv}(\mathbf{k}) = 2\pi\hbar\omega^2 \sum_m |p_{cm}^{TNL}|^2 |p_{mv}^{SPA}|^2 \delta\left( \epsilon + E_g - N\hbar\omega \right)$$

(14)

where the delta function demonstrates the energy conservation for the full transition. The transition rate $W_{cv}(\mathbf{k})$ is proportional to a product of the single-photon absorption probability $|p_{mv}^{SPA}|^2$ (electron promoted from the valence band to intermediate state) and the tunneling probability $|p_{cm}^{TNL}|^2$ (electron tunneling from intermediate state to conduction band). Expanding $|p_{mv}^{SPA}|^2$ in Eq. (14) in terms of higher-order multipoles[25], the absorption rate for left- and right-helical light (with identical polarization) represented by ± sign, $W_{cv}^{(\pm)}$, can be written as

$$W_{cv}^{(\pm)}(\mathbf{k}) = 2\pi \sum_m \left[ \langle |p_{cm}^{TNL}|^2 |\mu_i^{mv}|^2 \rangle_P |E_i^\pm|^2 + \langle |p_{cm}^{TNL}|^2 |m_i^{mv}|^2 \rangle_P |B_i^\pm|^2 + 2\langle |p_{cm}^{TNL}|^2 \mu_i^{mv} m_i^{vm} \rangle_P \right.$$
$$\left. \text{Im}\left[ E_i^{\pm*} B_i^\pm \right] + \frac{2}{3} \langle |p_{cm}^{TNL}|^2 \mu_i^{mv} \theta_{ij}^{vm} \rangle_P \text{Re}\left[ \nabla_i E_j^\pm E_i^{\pm*} \right] \right] \frac{\delta\left( \epsilon + E_g - N\hbar\omega \right)}{h(\omega_{mv} - \omega)^2}$$

(15)

where, $E$ and $B$ are incident electric and magnetic fields (Supplementary Section 7 for asymmetric Laguerre–Gaussian beam equation), $\mu_i$ represents the intrinsic electric dipole, $m_i$ is the intrinsic magnetic dipole, $\theta_{ij}$ represents the intrinsic electric quadrupole. Since photons are absorbed in an ordered environment within the short-range distances in both c-solids and a-solids, we assume that the anisotropic averaging, $\langle \rangle_P$, results in a finite orientation-dependent weighing factor

$\Omega_{i=1,4}$. As a result of this simplification, the above expression reduces to

$$W_{cv}^{(\pm)}(\mathbf{k}) = 2\pi \sum_m |p_{cm}^{TNL}|^2 \left[ \underbrace{\Omega_1 |\mu_i^{mv}|^2 |E_i^\pm|^2}_{\text{E1E1}} + \underbrace{\Omega_2 |m_i^{mv}|^2 |B_i^\pm|^2}_{\text{M1M1}} + \underbrace{\Omega_3 (2\mu_i^{mv} m_i^{vm}) \text{Im}\left[ E_i^{\pm*} B_i^\pm \right]}_{\text{E1M1}} \right.$$
$$\left. + \underbrace{\Omega_4 \left( \frac{2}{3} \mu_i^{mv} \theta_{ij}^{vm} \right) \text{Re}\left[ \nabla_i E_j^\pm E_i^{\pm*} \right]}_{\text{E1E2}} \right] \frac{\delta(\epsilon + E_g - N\hbar\omega)}{h(\omega_{mv} - \omega)^2}$$

(16)

where E1 and M1 are electric and magnetic dipoles, respectively, and E2 is the electric quadrupole. The coupling terms E1M1 and E1E2 are pseudoscalars and change signs under improper rotation. $W_{cv}^{(\pm)}(\mathbf{k})$ is a product of tunneling probability amplitude $|p_{cm}^{TNL}|^2$ and single-photon absorption probability amplitude $|p_{mv}^{SPA}|^2$. The above equation can be generalized to the multiphoton case and was also shown to be responsible for the origin of HD in chiral and achiral molecules[25].

HD (Type I) is proportional to the difference in the interband transition rates between left- and right-handed helical light with beam asymmetry parameter $\delta$ can be expressed as:

$$\Delta W_{cv} = W_\delta^+ - W_\delta^- = 2\pi \sum_m |p_{cm}^{TNL}|^2 \left[ \Omega_3 (2\mu_i^{mv} m_i^{vm}) \left( \text{Im}\left[ E_i^{+*} B_i^+ \right] - \text{Im}\left[ E_i^{-*} B_i^- \right] \right) \right.$$
$$\left. + \Omega_4 \left( \frac{2}{3} \mu_i^{mv} \theta_{ij}^{vm} \right) \left( \text{Re}\left[ \nabla_i E_j^+ E_i^{+*} \right] - \text{Re}\left[ \nabla_i E_j^- E_i^{-*} \right] \right) \right] \frac{\delta(\epsilon + E_g - N\hbar\omega)}{h(\omega_{mv} - \omega)^2}$$

(17)

E1E1 and M1M1 terms do not contribute to HD (Type I) because the field intensities and profiles remain the same for both helicities. Evaluation of above equation shows that HD (Type I) is a beam-dominated property as the material tensors are identical for both the helicities. Moreover, within dipole approximation, the tunneling contribution $|p_{cm}^{TNL}|^2$ results in identical rates for both helicities.

The HD (Type I) is dependent on the beam asymmetry parameter $\delta$ where $\delta = 0$ represents symmetric beam and $\delta \neq 0$ asymmetric beam (Supplementary Section 7 for beam profiles). The E1M1 contribution for both symmetric and asymmetric beam vanishes because we take the difference between the left- and right-helical beams for the same polarization. Therefore, HD (Type I) arises due to E1E2 coupling term. The E1E2 term contains the gradient of the electric field giving rise to $l$-dependence. Therefore, for symmetric beams E1E2 contributions average out to zero and only exist for asymmetric beams. For left- and right-circularly polarized light, the E1M1 term is nonzero and gives rise to conventional chiral signal

The total transition rate, $W_{cv}$, can be obtained by integrating, $W_{cv}(\mathbf{k})$, over all momentum states. For numerical estimation, shown in Fig. 5, we assume the summation over all intermediate states in the above equation is dominated by a single state that is pre-determined by the incident photon. The total transition rate is enhanced when the intermediate state is in resonance with the incident photon energy. In addition, we use linearly polarized light to simplify $W_{cv}(\mathbf{k})$ and perform the integration using the properties of Bessel's functions and the weak-field limit for the non-perturbative discrete transition rate[31,62,63] (Supplementary Section 9 for details) to obtain

$$\Delta W = W_\delta^+ - W_\delta^- = \frac{2\Omega_4(\mu_i^{mv} \theta_{ij}^{vm})}{3h^2(\omega_{mv} - \omega)^2} \widetilde{W}\left( \text{Re}\left[ \nabla_i E_j^+ E_i^{+*} \right] - \text{Re}\left[ \nabla_i E_j^- E_i^{-*} \right] \right) = D(\Upsilon^+ - \Upsilon^-)$$

(18)

where $\Upsilon^\pm = \text{Re}\left[ \nabla_i E_j^\pm E_i^{\pm*} \right]$ represent the optical helicity describing the handedness of the helical light, its association with the field gradient gives rise to $l$-dependence. This quantity is odd under parity with a change in the sign of the displacement of the singularity, $\delta$, and time

reversal changes the handedness of helical light. Also,

$$\tilde{W} = \frac{1}{(2\pi)^4 \hbar} \left(\frac{m_1 \omega p_{cm}}{m(\eta-1)!}\right)^2 \left(\frac{eA_0}{2m_1\omega}\right)^{2\eta} \left(\frac{2m^*}{\hbar^2}\right)^{(2\eta+1)/2} \left(\frac{(n\hbar\omega - E_g)}{(2\eta-1)}\right)^{(2\eta-1)/2} \quad (19)$$

where $n = \eta + 1$ is the total photons absorbed for linearly polarized light. HD(Type I) can now be defined in terms of energy absorbed $\Gamma$ normalized with respect to the incident laser energy, $\mathcal{E}_{inc}$ as:

$$\Delta\boldsymbol{\Gamma} = \Gamma^+ - \Gamma^- = \frac{\hbar(\mathsf{W}_\delta^+ - \mathsf{W}_\delta^-)}{\mathcal{E}_{inc}} = \mathcal{D}(\Upsilon^+ - \Upsilon^-) \quad (20)$$

where $\mathcal{D} = \frac{\hbar \mathbf{D}}{\mathcal{E}_{inc}}$. HD (Type I) is obtained by integrating over the beam cross-section $\int_{-w_0}^{w_0} \Delta\boldsymbol{\Gamma} \, dx dy$

For an order of magnitude estimation of $\mathcal{D}$, we used the following values; the intrinsic dipole as $\mu_i \approx 1.65 \times 10^{-30}$ Cm[65], the quadrupole for fused silica molecule as $\theta_{ij} \approx 9.3 \times 10^{-49}$ Cm²(glass)[66], $A_0 = E_0/\omega$ where $E_0 = 7.1 \times 10^9$ to $1.5 \times 10^{10}$ V/m and $\omega = 2.35 \times 10^{15}$ Hz. $p_{cm}/m \approx E_g'/m^{*}$[67]$E_g'$ is the bandgap between intermediate and conduction band (-7.4 eV), $m^* = m_0 = m_1 \approx 10^{-30}$ kg[34], and $E_g \approx 9$ eV is the full bandgap. We assumed (i) $\Omega_4 = 1$ (ordered environment for crystals and amorphous materials within the short-medium-range distances), (ii) single level dominates the total transition rate, (iii) The summation of the Bessel functions is evaluated by retaining in the sum over n only the term closest to resonance for the intermediate state tunneling contribution and (iv) one-photon transition is highly nonresonant so that $\omega_{mv} - \omega \approx \omega$[64] (v) dominant component of optical helicity ($\Upsilon$) with respect to displacement of singularity. Due to discretization of transition from intermediate state to the conduction band during the tunneling process we assume $\eta = 5$ and $n = \eta + 1$ corresponding to bandgap energy. Thus, the value of $\mathcal{D}$ is $7.3 \times 10^{-5}$. The above numerical estimate was plotted in Fig. 5a, c as a function of displacement of singularity. The estimated HD values differ by few orders of magnitude compared to experimental values. This variation could be due the approximate values used for electric dipole and quadrupole moment and the material response.

For n-photon transitions, $|p_{mv}^{SPA}|^2$ should be substituted by the multiphoton contribution ($|p_{mv}^{MPA}|^2$) in the transition rate $W_{cv}(\mathbf{k})$ and will contain multitude of cross-correlation terms arising from the summation in the modulus square of the transition amplitude. However, optical helicity $\Upsilon^\pm$ will still be present giving rise to the observed HD (Type I)[25].

Helical dichroism obtained using MPAT is not only limited to the tunneling interaction Hamiltonian $\hat{V}_f$ written in the dipole approximation. Our experimental results can also be described as long as one or both of the interaction Hamiltonian either $\hat{V}_f$ ($\hat{V}_i$) used in the tunneling (multiphoton) probability amplitude includes the contribution of higher-order multipoles to obtain the E1E2 coupling term. For higher incident intensities, the interband excitations are dominated by ground-state tunneling transitions. In this regime, the HD can be explained as long as the interaction Hamiltonian contains the contribution of higher-order multipoles which gives rise to the $l$-dependence.

## Superposition of OAM and Gaussian

The experimental optical setup consists of a combination of waveplates and a $q$-plate (Supplementary Section 1). When an incident circularly polarized Gaussian beam propagates through the $q$-plate, it acquires an OAM defined by $l = \pm 2q$. Experimentally, we can measure the efficiency of the the conversion process. If the conversion efficiency is not 100%, as is often the case, the transmitted light will consists of a superposition of Gaussian and Laguerre–Gaussian beam. In other words, a portion of the incident Gaussian will transmit

unaffected and the remaining portion will be converted to OAM. In our setup, the incident light is vertically polarized.

$$|V\rangle \xrightarrow{\text{QWP}(\pm 45)} \overset{\text{LHC}}{\longrightarrow}$$

$$\text{RHCQ-plate} \overset{l+}{\longrightarrow} \quad (21)$$

$$l\text{-QWP}(\mp 45) \longrightarrow \text{QWP}(\theta)$$

Vertically polarized light incident onto the first quarter-wave plate (QWP) generates left-CPL (Right-CPL) if the angle of incidence is 45° (−45°) with respect to the horizontal axis. Experimentally, the transmitted light has an ellipticity between 90 and 95% therefore the major and minor axes are not identical. We must take this in account by introducing two variables $\alpha$ and $\beta$ (such that $|\alpha|^2 + |\beta|^2 = 1$) which are used to vary the ellipticity of the transmitted light that is subsequently incident on the q-plate. Depending on the conversion efficiency of the $q$-plate, a portion of the light will remain unchanged and the other will acquire an $l$-value with opposite polarization due to the retardance of $\pi$ radians. The ratio of Gaussian to Laguerre–Gaussian is determined by the retardation angle $\xi$. The second QWP was used to generate linearly polarized superimposed Gaussian and OAM beam. The third QWP could be used to produce the desired ellipticities. To determine the polarization state after each optics we will implement the Jones matrices convention for QWP($\theta$) and HWP($\theta$), where $\theta$ is the angle with respect to the horizontal axis[68,69]. We can define the operation of the q-plate by means of a unitary operator $M_Q$[57,70]. The handedness of the helical light is determined by the incident circular polarization on the q-plate.

$$\begin{cases} M_Q|\text{LHC}\rangle = \cos\left(\frac{\xi}{2}\right)|\text{LHC}\rangle + i\sin\left(\frac{\xi}{2}\right)|\text{RHC}\rangle e^{il\phi} \\ M_Q|\text{RHC}\rangle = \cos\left(\frac{\xi}{2}\right)|\text{RHC}\rangle + i\sin\left(\frac{\xi}{2}\right)|\text{LHC}\rangle e^{-il\phi} \end{cases} \quad (22)$$

The portion of the light that acquires an $l$-value also gains a $\pi$ phase shift between polarization components i.e., the q-plate acts like a HWP with respect to the incident polarization. The output unit vectors of the electromagnetic field can be determined via Jones matrix multiplication of the second and third QWP. Introducing the spatial components of the incident light we can express the electric field for both helicities as

$$\begin{pmatrix} E_x \\ E_y \end{pmatrix}_{l\pm} = \frac{1}{\sqrt{2}} \left[ \cos(\xi/2)\begin{pmatrix} \alpha \\ i\beta \end{pmatrix} e^{-\frac{i\pi}{2}} u_g(x,y,z) + \sin(\xi/2)\begin{pmatrix} \beta \\ -i\alpha \end{pmatrix} u_0^\pm(x,y,z) \right] \quad (23)$$

The magnetic field can be obtained via the relation $\mathbf{B}(x,y,z) = \hat{z} \times \mathbf{E}(x,y,z)$.

$$\begin{pmatrix} B_x \\ B_y \end{pmatrix}_{l\pm} = \frac{1}{\sqrt{2}} \left[ \cos(\xi/2)\begin{pmatrix} \beta \\ i\alpha \end{pmatrix} e^{-i\pi} u_g(x,y,z) + \sin(\xi/2)\begin{pmatrix} i\alpha \\ \beta \end{pmatrix} u_0^\pm(x,y,z) \right] \quad (24)$$

These expressions were used for the simulations shown in Fig. 5a, c. The $u_0^\pm(x,y,z)$ and $u_g(x,y,z)$ are the Laguerre–Gaussian and Gaussian beam expressions given in Supplementary Section 7[25]. The $\pm$ sign represents the handedness of the helical light.

## Data availability

The minimum dataset necessary to interpret the results can be obtained from the corresponding authors upon request. The raw and processed data are not deposited in a repository because transmission measurements generate multiple sets of columns and without proper context the data could be hard to interpret.

## Code availability

The simulation data were obtained by evaluating the equations using standard technical software. The code is available upon request to the corresponding authors.

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

## Acknowledgements

The authors thank PhD student Felix Hufnagel from Prof. Karimi's group for fabricating the q-plate used in the experiments. We also acknowledge financial support from the Natural Science and Engineering Council of Canada, Canada Research Chairs, and Canadian Foundation for Innovation.

## Author contributions

A.J., J.-L.B., and R.B. conceived, designed, and planned the experiments. A.J. and J.-L.B. conducted the experiments and analyzed the results. A.J., J.-L.B., T.B., and R.B. worked on the theory and conducted numerical simulations. P.C., E.K., and R.B. supervised the project. A.J., J.-L.B., T.B., and R.B. prepared the first draft, and all authors reviewed the manuscript.

## Competing interests

The authors declare no competing interests.
