## [Peer Review File · Nature Communications]

Intrinsic dichroism in amorphous and crystalline solids with helical lightReviewer #1 (Remarks to the Author):

This is a relatively interesting manuscript, where the authors have taken a methodology they first published in Nature Photonics earlier this year to study amorphous solids with vortex beams. The physics and experimental plots produced in the manuscript seems correct to me. The biggest issue I have with this work is that it isn't explained at all well, nor has it been placed into much context. It is my opinion the manuscript needs a thorough re-write to bring the clarity of the text up to a standard one would expect in Nature Communications. My biggest issue stems with their definition of Type 1 and Type 2 dichroism (see below). Work in the field of chirality of structured light and its interaction with matter is a hot topic at the moment, so I believe the work will have suitable impact for Nat Comms (though note the authors have already published a similar type of work). After the clarity of their manuscript is improved I would support publication.

Specific comments:

Helical light beams as a phrase is ambiguous, so is 'helical light'. The reason is that circularly polarized light has a helical (electromagnetic field) structure. The authors here are of course concerned with light that has a helical wavefront (surface of constant phase), best known as optical vortices. Interchanging the word helical with vortex removes any ambiguity. In the literature many use vortex rather than helical, but some authors have unfortunately decided to use helical in this context. Anyone skimming this manuscript would be forgiven if they thought the work was about the helical field vectors of circularly polarised light, rather than the helical structure of the wavefront. I am not requesting the authors change their phrases, but they will be perpetuating this misnomer as it stands.

No need to capitalise 'circular polarised light', nor 'circular dichroism'.

On Page 2 can the authors give a more thorough definition of their Type 1 and Type 2 'HD', I find them a little ambiguous as given here. I think Type 1 is any material but differential absorption of chiral light; Type 2 is a chiral material given differential absorption irrespective of the chirality of the light? This raises some interesting symmetry issues before one gets into the details. It is rather obvious that, on symmetry grounds alone, these interactions in this work (under these definition given by the authors) are not chiral effects in the traditional sense of being represented by space odd, time even pseudoscalars. I think it is extremely important the authors point this out at this early stage. For observation of true chirality (natural optical activity) you require chiral light and chiral material. In this work it looks to me like the authors are simply exposing materials to a differing intensity distribution of light, in which case it is rather obvious that more/less light will be absorbed/scattered as the distribution of light is altered upon a material locked in place – it is like turning a light on and off at any given local position. Let me state it is still very interesting that this effect depends on the sign of the topological charge! I just feel like the phenomena they have discovered should be placed in the correct context.

----- I come back to the above comment after reading the full manuscript and note that, for example, Figure 3 highlights both Type 1 and Type 2 which depend on both material and optical chirality. This is what I'd expect for a true optical activity effect. However, the definitions given in the Introduction do not suggest this is what is going to be achieved. Essentially, I believe the authors should really spend some time in the Introduction clarifying their Type 1 and Type 2 mechanism with explicit reference to spatial parity and time parity symmetry signatures.

The statement on Page 3: 'Dichroism does not exist in achiral solids for a Gaussian beam and also for a symmetric OAM beam' needs significant clarification. What dichroism are the authors concerned with? 'Helical', circular, linear? What is stopping linear dichroism existing in this scenario, or circular dichroism if your Gaussian beam was circularly polarized?

Page 6 – 'equations 17-20'?

It is unfortunate that the authors have failed to reference any of the previous literature (other than their own) on chiral interactions of vortex beams/structured light with matter in the main manuscript, e.g., the studies which discovered the importance of the quadrupole, orientational

order etc. This would place their work in more context than it currently is.

Reviewer #2 (Remarks to the Author):

In this manuscript, Ashish Jain et al. report the discovery of two types of helical dichroism in intense laser-solid interaction experiments. First, they conducted experiments with asymmetric helical, linearly polarized, 100-fs laser beams at 800 nm. They observed large difference in the transmission of left and right asymmetric helical beams through 1-mm thick samples when the laser fluence is above a certain level. Such type I helical dichroism was found in both crystalline and amorphous solids no matter the materials are chiral or not. Then they discovered that chiral crystals demonstrate another type of helical dichroism (type II) that is more efficient than the conventional circular dichroism. They developed a theoretical model based on the multiphoton assisted tunneling (MPAT) excitation, which qualitatively reproduced some of the experimental results. The research work is original and exciting. They added another dimension to the understanding of strong field physics in condensed matter. The discovery paved new ways to measure the structure, symmetry, and chirality of crystals with high sensitivity. I can recommend publishing the paper in Nature Communications if the following issues are clarified.

1. For amorphous solids, what properties (size of the domain?) can be measured by the helical dichroism?
2. When the Keldysh parameters are in the range of ~ 1.1 to 2.5 , the Perelomov-Popov-Terent'ev (PPT) model should work since it includes both multiphoton and tunnel effects. What is the reason that it is not applied and the MPAT is used instead? What is the difference between the two models?
3. Would the helical dichroism be stronger with lasers at longer wavelengths since the Keldysh parameters are smaller?

Response to Reviewer #1:

Thank you for positive response to our work. Below is the response to your comments and the changes made to improve the delivery of our research. Below we respond to your comments in detail. (Comments in red, responses in blue).

General comment:

This is a relatively interesting manuscript, where the authors have taken a methodology they first published in Nature Photonics earlier this year to study amorphous solids with vortex beams. The physics and experimental plots produced in the manuscript seems correct to me. The biggest issue I have this work is that it isn't explained at all well, nor has it been placed into much context. It is my opinion the manuscript needs a thorough re-write to bring the clarity of the text up to a standard one would expect in Nature Communications. My biggest issue stems with their definition of Type 1 and Type 2 dichroism (see below). Work in the field of chirality of structured light and its interaction with matter is a hot topic at the moment, so I believe the work will have suitable impact for Nat Comms (though note the authors have already published a similar type of work). After the clarity of their manuscript is improved I would support publication.

HD (Type I) is evaluated by taking the difference in absorption between the opposite handedness of the helical phase for the same material. It is a beam dominated property and its definition is independent of the material symmetry. HD (Type I) exists in amorphous, achiral and chiral crystalline solids.

HD (Type II) is evaluated by taking the difference of absorption between the left- and right-handed chiral solids for a specific handed helical light. It is a material dominated property and its definition involves both the material and light beam to be chiral entities.

To emphasize, the main objectives of the manuscript are as follows.

First, demonstrate existence of intrinsic dichroism in amorphous solids using asymmetrical helical phase-based light beams that carry orbital angular momentum (OAM). In general, any dichroism in amorphous material (differential absorption of either polarized or helical phase-based light beams) is not expected to be observed due to the disorderly state of the medium.

Second, demonstrate high chiral sensitivity compared to other existing solid-state based CD techniques to study chiral matter. Efficiency of HD was an order of magnitude higher than the reported values of CD in complex chiral crystals [Phys. Rev. Lett. 80, 21 (1998)]. There is dearth of information on the magnitude of chiral signal in quartz possibly due to birefringence induced errors. Recent experimental studies on HD in powdered molecular media using x-rays obtained comparable efficiency [Nat. Photon. 16, 570–574 (2022)] while theoretical studies have only qualitatively implied existence of phase-based dichroism in orientated materials [Phys. Rev. A. 99, 023837 (2019), Optics Letters 43, 3435–438 (2018)].

Third, the origin of such helicity-based dichroism was presented in terms of inter band transitions that are facilitated by multiphoton assisted electron tunneling (excited state tunneling transitions) which is within the short-range order leading to non-zero HD signal in amorphous materials.

Fourth, we demonstrate precise control and tuning of HD signal by varying the superposition of OAM and Gaussian beams. This combined with the bandgap dependence of HD are unique features our technique.

To make our work more accessible to the reader we modified the introduction to include (a) the properties of OAM beams, (b) earlier studies on phase based dichroism, (c) expanded definition of HD (Type I, II), (d) PT symmetry, and (e) the context of the main results.

Below we respond to your specific comments.

Specific comments:

(1) - Helical light beams as a phrase is ambiguous, so is 'helical light'. The reason is that circularly polarized light has a helical (electromagnetic field) structure. The authors here are of course concerned with light that has a helical wavefront (surface of constant phase), best known as optical vortices. Interchanging the word helical with vortex removes any ambiguity. In the literature many use vortex rather than helical, but some authors have unfortunately decided to use helical in this context. Anyone skimming this manuscript would be forgiven if they thought the work was about the helical field vectors of circularly polarised light, rather than the helical structure of the wavefront. I am not requesting the authors change their phrases, but they will be perpetuating this misnomer as it stands.

We acknowledge that helicity can also refer to the structure of electric field in case of circularly polarised light and can be confused with the helical structure of wavefront associated with optical vortex beams. We now state this clearly in the modified introduction where a brief discussion on OAM beams is presented. Our labeling of helically phased optical vortex beams as a "helical light beams" is based on several publications and books [*Opt. Commun.* 237, (2004)]. Moreover, authors of previous articles in the last decade coined the term Helical Dichroism (HD) to describe differential absorption between left and right-handed vortex beams [*Sci. Adv.* 2, 150134 (2016), *ACS Nano* 15,2893–2900 (2021)]. To keep our work consistent with past articles, we continued the use of the term HD. However, to clarify and not to create any unintentional misnomer, we have added the discussion on helical light beams where helicity refers to the phase of the beam. Such a clarification should be helpful to the potential reader.

(3) - No need to capitalise 'circular polarised light', nor 'circular dichroism'.

Thanks for pointing it out. We have corrected it wherever applicable.

(4a) - On Page 2 can the authors give a more thorough definition of their Type 1 and Type 2 'HD', I find them a little ambiguous as given here. I think Type 1 is any material but differential absorption of chiral light; Type 2 is a chiral material given differential absorption irrespective of the chirality of the light?

HD(Type I) is defined as the difference in absorption of left- and right-handed helical light irrespective of material symmetry. When a chiral solid is involved in the interaction with chiral light, HD(Type I) represents true chirality and is analogous to definition to CD whereas in achiral solids it simply represents a difference in absorption of left- and right-handed helical light. In contrast, HD(Type II) is defined as the difference in absorption between left- and right-handed chiral solid for a specific handedness of the helical light. This is valid only for chiral light interaction with chiral matter. These differences between the two types of HD are now clearly highlighted in the introduction.

(4b) This raises some interesting symmetry issues before one gets into the details. It is rather obvious that, on symmetry grounds alone, these interactions in this work (under these definition given by the authors) are not chiral effects in the traditional sense of being represented by space odd, time even pseudoscalars. I think it is extremely important the authors point this out at this early stage. For observation of true chirality (natural optical activity) you require chiral light and chiral material.

Our results were described by considering multipole interaction where two main coupling terms are relevant.

Conventional CD is described by the electric-magnetic dipole coupling term $E1M1 \propto \langle \mu_\alpha^{mg} m_\alpha^{gm} \rangle C^\pm$ where C represents optical chirality defined as $C^\pm = 2\omega \text{Im}[(\tilde{E}_\delta^\pm)^* \cdot \tilde{B}_\delta^\pm]$. This coupling term is a space odd - time even pseudoscalar and can be shown as

$$PT \left\{ \frac{\omega}{2} \text{Im}[E^* \cdot B] \right\} \rightarrow P \left\{ -\frac{\omega}{2} \text{Im}[E \cdot B^*] \right\} \rightarrow P \left\{ \frac{\omega}{2} \text{Im}[E^* \cdot B] \right\} = -\frac{\omega}{2} \text{Im}[E^* \cdot B]$$

Therefore, optical chirality is odd under PT symmetry.

HD (Type I, II) is described by the electric dipole-quadrupole coupling term $E1E2 \propto \langle \mu_\alpha^{mg} \theta_{\alpha\beta}^{gm} \rangle Y^\pm$ where optical helicity Y is defined as $Y^\pm = \text{Re}[\nabla_i E_j^\pm E_i^{\pm*}]$. The optical helicity is a vector quantity that is odd under parity with a change in the sign of the displacement of the singularity, δ , and time reversal only changes the handedness of helical light, as shown below. Also, optical helicity is polarization invariant.

The electric field is written in cartesian coordinates, where the handedness is expressed as $(x \pm iy)^{|l|} = r^{|l|} e^{\pm i l \phi}$. Asymmetrical LG modes are given as $E(\pm l, \pm \delta) = [(x \mp i\delta) \pm i(y \mp i\delta)]^{|l|} = [(x + \delta) \pm i(y - \delta)]^{|l|}$ (& $E(\mp l, \pm \delta) = [(x + \delta) \mp i(y - \delta)]^{|l|} = E(\pm l, \pm \delta)^*$) and using following properties:

$$\begin{aligned} P(E(\pm l, \pm \delta)) &\rightarrow P([(x \mp i\delta) \pm i(y \mp i\delta)]^{|l|}) \rightarrow (-1)^l [(x \pm i\delta) \pm i(y \pm i\delta)]^{|l|} \\ &\rightarrow (-1)^l [(x - \delta) \pm i(y + \delta)]^{|l|} \rightarrow (-1)^l E(\pm l, \mp \delta) \end{aligned}$$

If polarization is considered, then $(-1)^l$ will become $(-1)^{l+1}$ and

$$\begin{aligned} T(E(\pm l, \pm \delta)) &\rightarrow T([(x \mp i\delta) \pm i(y \mp i\delta)]^{|l|}) \rightarrow [(x \pm i\delta) \mp i(y \pm i\delta)]^{|l|} \\ &\rightarrow [(x + \delta) \mp i(y - \delta)]^{|l|} \rightarrow E(\mp l, \pm \delta) \end{aligned}$$

Therefore, the optical helicity under PT symmetry is given as

$$\begin{aligned} PT[Y_{\pm\delta}^\pm] &\rightarrow PT \left[\text{Re}[\nabla_i E_j(\pm l, \pm \delta) E_i(\pm l, \pm \delta)^*] \right] \rightarrow P \left[\text{Re}[\nabla_i E_j(\pm l, \pm \delta) * E_i(\pm l, \pm \delta)] \right] \rightarrow \\ &P \left[\text{Re}[\nabla_i E_j(\mp l, \pm \delta) E_i(\mp l, \pm \delta)^*] \right] \rightarrow -\text{Re}[\nabla_i E_j(\mp l, \mp \delta) E_i(\mp l, \mp \delta)^*] = -Y_{\mp\delta}^\mp \end{aligned}$$

Without showing all the above details in the manuscript, we now simply point out the parity and time symmetry of optical helicity.

(4c) In this work it looks to me like the authors are simply exposing materials to a differing intensity distribution of light, in which case it is rather obvious that more/less light will be absorbed/scattered as the distribution of light is altered upon a material locked in place – it is like turning a light on and off at any given local position. Let me state it is still very interesting that this effect depends on the sign of the topological charge! I just feel like the phenomena they have discovered should be placed in the correct context.

We believe your comments about the obviousness of the results appear to arise from a misunderstanding of how HD was measured experimentally and evaluated numerically. We would like to point out clearly that since HD signal is averaged over the entire beam cross-section, therefore, we are not considering the localized effects.

Experimentally, HD was obtained by (a) focussing asymmetric optical vortex (singularity displaced) beams with opposite helicity inside the bulk solid sample (amorphous and crystalline) and (b)

collecting the whole beam onto a photodiode to obtain transmission/absorption signal over the interaction region. The average intensity for all positions of the displaced singularity and different helicity ($\pm l$ - value) remains same throughout experiment (within instrumental errors).

Theoretically, the HD simulated in fig 5, was obtained by integrating over the beam cross section (spatial averaging). This was mentioned in the text on page 7 (in paragraph after eq. 1). For symmetrical OAM beams (singularity at center), the spatial averaging cancels any observed differences between the left- and right-handed helical beams.

To clarify further, the mathematical expression is explicitly shown in fig. 5 caption and in the text on page 6.

(5) - ----- I come back to the above comment after reading the full manuscript and note that, for example, Figure 3 highlights both Type 1 and Type 2 which depend on both material and optical chirality. This is what I'd expect for a true optical activity effect. However, the definitions given in the Introduction do not suggest this is what is going to be achieved. Essentially, I believe the authors should really spend some time in the Introduction clarifying their Type 1 and Type 2 mechanism with explicit reference to spatial parity and time parity symmetry signatures.

We agree that more context has to be added to the introduction to elaborate on HD(Type I, II). PT symmetry can provide information on broken symmetry in specific light-matter interactions and natural optical activity. However, it cannot provide complete insight into the existence of intrinsic HD in amorphous solids. We introduced MPAT model to describe the existence of HD in amorphous and crystalline solids. MPAT describes the following behaviour of our results which cannot be obtained solely by PT symmetry arguments.

- The variation of HD(Type I) with displacement of singularity (sinusoidal behaviour)
- The tunability of HD(Type I) with OAM-Gaussian superposition .
- The effect of ellipticity on HD(Type I).
- The influence of material bandgap on the magnitude of HD(Type I).

This is the reason why symmetry arguments were not mentioned in the manuscript. Nonetheless, we understand that PT arguments are a traditional approach to describe chiral light-chiral matter interactions, therefore, below we elaborate on the definitions of HD(Type I, II) in the context of parity-time symmetry properties.

PT symmetry on HD(Type I)

Using property: $PT[Y_{\pm\delta}^{\pm}] \rightarrow -Y_{\mp\delta}^{\mp}$

Case a- Achiral system (interaction with asymmetrical left- and right-handed helical beams): According to equation 1 in the article, (beam helicity is changed; symmetrical or asymmetrical?)

$PT [HD(\text{Type I})] = PT[\mathcal{D} [Y_{+\delta}^+ - Y_{+\delta}^-]] = \{PT [\mathcal{D}]\} [-Y_{-\delta}^- + Y_{-\delta}^+] = -\mathcal{D} [Y_{-\delta}^+ - Y_{-\delta}^-]$ where $PT[\mathcal{D}] = PT\left[\frac{2\Omega_a(\mu_i^{mv}\theta_{ij}^{vm})}{3\hbar\epsilon_{inc}(\omega_{mv}-\omega)^2}\tilde{W}\right] = -\mathcal{D}$, since electric dipole transition moment is space odd-time even and quadrupole is space even-time even transition moment. Here \mathcal{D} includes the multiphoton and tunneling components of the interband transitions. Therefore HD (Type I) is odd under PT symmetry with respect to $\pm\delta$ position. This is observed in our experimental results (Figs. 1,2 & 4,5) with achiral solids where the sign of HD changes with respect to $\pm\delta$ position.

Case b- Chiral system (interaction with asymmetrical left- and right-handed helical beams for a fixed handedness of chiral solid): For simplicity, we draw parallels between enantiomers and chiral solids. PT symmetry on R-enantiomer will lead to S-enantiomer $PT[\mathcal{D}_R] = -[\mathcal{D}_S]$. This can be shown by assigning the symmetrical ground state Ψ_s to the R-enantiomer and the anti-symmetrical Ψ_{as} to the S-enantiomer. This can be justified as the consequence of the slight asymmetry in the molecular potential, leading to eigenfunctions either symmetrical $\Psi_s^g = \frac{1}{\sqrt{2}} (\psi_+ + \psi_-)$ or anti-symmetrical $\Psi_{as}^g = \frac{1}{\sqrt{2}} (\psi_+ - \psi_-)$, where Ψ_+ (Ψ_-) is even (odd) under parity. Therefore, the enantiomer pair is localized in opposite ground states; if R is even, then S is odd or vice versa. The excited states for both enantiomers are assumed to be a mixed parity state given as $\Psi_m = \frac{1}{\sqrt{2}} (\psi_+ + i\beta\psi_-)$, where $i\beta$ is a small phase lag factor between even and odd states [G. Wagnière, (John Wiley and Sons, 2007), Phys. Rev. A 105, 022803 (2022)., J. Chem. Phys. 159, 014504 (2023)]. Therefore,

$$PT [\text{HD}(\text{Type I})] = PT[\mathcal{D}_R [Y_{+\delta}^+ - Y_{+\delta}^-]] = \{PT [\mathcal{D}_R]\} [-Y_{-\delta}^- + Y_{-\delta}^+] = -\mathcal{D}_S [Y_{-\delta}^+ - Y_{-\delta}^-]$$

This is observed in our experimental results (Fig. 3a) with chiral solids where the sign of HD changes with respect to δ position for each enantiomer. In this case, HD(Type I) can be considered as true chirality and is analogous to definition of CD.

PT symmetry on HD(Type II) (interaction of left- and right-handed chiral solid with a fixed handedness of helical light):

For asymmetrical OAM beams:

$$PT\{HD(\text{Type II})\} = \frac{PT\{R(\pm l, \pm \delta) - S(\pm l, \pm \delta)\}}{PT\{R(\pm l, \pm \delta) + S(\pm l, \pm \delta)\}}$$

$$PT\{HD(\text{Type II})\} = \frac{P\{R(\mp l, \pm \delta) - S(\mp l, \pm \delta)\}}{P\{R(\mp l, \pm \delta) + S(\mp l, \pm \delta)\}}$$

$$PT\{HD(\text{Type II})\} = \frac{-S(\mp l, \mp \delta) + R(\mp l, \mp \delta)}{-S(\mp l, \mp \delta) - R(\mp l, \mp \delta)}$$

$$PT\{HD(\text{Type II})\} = -\frac{R(\mp l, \mp \delta) - S(\mp l, \mp \delta)}{R(\mp l, \mp \delta) + S(\mp l, \mp \delta)}$$

Therefore, HD(Type II) is odd under time-parity symmetry and is also observed in our experimental results presented in Fig. 3b. This could be considered “true chirality”. Without showing all the above details in the manuscript, we now point out the explicit parity and time symmetry of HD(Type I) and HD(Type II).

(6) - The statement on Page 3: ‘Dichroism does not exist in achiral solids for a Gaussian beam and also for a symmetric OAM beam’ needs significant clarification. What dichroism are the authors concerned with? ‘Helical’, circular, linear? What is stopping linear dichroism existing in this scenario, or circular dichroism if your Gaussian beam was circularly polarized?

Experimentally, dichroism (both circular and linear) was not observed for Gaussian beams as shown below in achiral MgO crystal. For symmetrical OAM beam, irrespective of polarization, HD(Type I) does not exist as shown in Fig.1 The statement on page 3 emphasizes this fact.

(7) - Page 6 – ‘equations 17-20’?

Here we were referring to set of equations from 17 to 20 used in deriving equation 1 - HD (Type-I). We have changed it just eq 20 showing the derived result.

(8) It is unfortunate that the authors have failed to reference any of the previous literature (other than their own) on chiral interactions of vortex beams/structured light with matter in the main manuscript, e.g., the studies which discovered the importance of the quadrupole, orientational order etc. This would place their work in more context than it currently is.

We have modified the introduction to discuss optical vortex beams carrying a helical phase front and the importance of previous studies on chiral light matter interaction. Here we provided additional references to previous work.

Response to Reviewer #2:

Thank you for positive evaluation of our work and recognizing its importance in providing new insights into light-matter interactions. Below we respond to your comments in detail.
(Comments in red, responses in blue).

In this manuscript, Ashish Jain et al. report the discovery of two types of helical dichroism in intense laser-solid interaction experiments. First, they conducted experiments with asymmetric helical, linearly polarized, 100-fs laser beams at 800 nm. They observed large difference in the transmission of left and right asymmetric helical beams through 1-mm thick samples when the laser fluence is above a certain level. Such type I helical dichroism was found in both crystalline and amorphous solids no matter the materials are chiral or not. Then they discovered that chiral crystals demonstrate another type of helical dichroism (type II) that is more efficient than the conventional circular dichroism. They developed a theoretical model based on the multiphoton assisted tunneling (MPAT) excitation, which qualitatively reproduced some of the experimental results. The research work is original and exciting. They added another dimension to the understanding of strong field physics in condensed matter. The discovery paved new ways to measure the structure, symmetry, and chirality of crystals with high sensitivity. I can recommend publishing the paper in Nature Communications if the following issues are clarified.

(1) For amorphous solids, what properties (size of the domain?) can be measured by the helical dichroism?

We believe HD measurements in amorphous solids could provide information on the following properties.

(a) HD in amorphous solids could, in principle, provide an estimate to the upper limit of short-range order. This is because in MPAT process, the tunneling displacement (x_0) is comparable to short-medium range order. For example, in fused silica it is roughly $\sim 20 \text{ \AA}$. As a result, isotropic averaging is restricted to an ordered environment giving rise to finite HD. The x_0 (eq. 2, main text) is inversely proportional to the incident field strength, therefore, at low intensities it becomes larger

than short range order and isotropic averaging over disordered domain will result in vanishing of higher order multipole moments, which leads to zero HD signal.

The following figure shows normalized transmission vs incident intensity in fused silica (disordered medium) and crystalline quartz (ordered medium). HD is the difference between the opposite helicity curves for respective materials. Since the nonlinear thresholds are dependent on the bandgap of the material, both materials should have similar onset since their bandgaps are comparable ($\sim 9.0 - 9.5$ eV). However, the onset for fused silica is at higher intensity compared to quartz (see figure). This difference could be attributed to electron displacement being greater than the short-range order, since x_0 is inversely proportional to intensity. This could, in principle provide an upper limit of short-range order. Estimating short-range order by HD measurements needs further investigation with focus on precise calibration (minimizing the experimental errors) of nonlinear thresholds in amorphous and crystalline solids.

HD in amorphous materials cannot measure the domain size but can only serve as an estimate of short-range order since the x_0 is calculated based on semi-classical approach to the quantum interaction containing several simplifying approximations.

(b) The bandgap dependence of HD shown in Fig. 4 (main text) enables to explore HD as a potential tool for material profiling. HD signal can serve as a unique fingerprint signature of the material whose amplitude varies with the band gap. It can also be used as an *in-situ* tool to study locally induced structural changes (for example, ultrafast laser modification of materials) on a micron scale inside a solid.

(c) HD in dielectrics could potentially aid research in ultrafast signal processing [Nature, 493, 70–74 (2013)] by enabling differential turn-on and turn-off of incident signals using asymmetric helical light as a control.

(2) When the Keldysh parameters are in the range of ~ 1.1 to 2.5 , the Perelomov-Popov-Terent'ev (PPT) model should work since it includes both multiphoton and tunnel effects. What is the reason that it is not applied and the MPAT is used instead? What is the difference between the two models?

Nonlinear inter-band transitions in solids can occur through two quantum pathways typically categorized as multiphoton and ground-state tunneling. The dominance of a particular pathway is often determined by the incident light intensity. When light intensity is in an intermediate region (Keldysh parameter $\sim 1-2$) where the above pathways are not dominant by themselves, a third pathway (known as MPAT) plays a prominent role. MPAT was originally introduced for gas phase ionization. In our article we extended this model to solids where HD was observed only at intermediate intensities. The following are the differences between the PPT and MPAT models.

PPT is a single-bound state theory of photoionization and does not fully include the role of bound excited states. In contrast, MPAT includes the excited state dynamics in which the electron can tunnel from the excited state to the continuum within a fraction of the field cycle. This new pathway channel was shown to dominate the dynamic leakage of electron wavepacket into the continuum over the ground-state tunneling or multiphoton ionization/excitation.

In PPT, tunnel ionization of atoms/molecules is described via dipole dominant interaction which should result in similar ionization probabilities for both helicities (without considering

the spatial profile of the fields). However, the origin of HD was shown to be arising from the higher order multipole interaction term which were considered in our extension of MPAT.

The PPT model considers the field strength and inadvertently neglects the transverse spatial structure of the beam. Since HD arises only in the case of asymmetrical Laguerre gaussian beams, therefore, the spatial transverse structure is vital to the observation and control of HD which is considered in MPAT.

(3) Would the helical dichroism be stronger with lasers at longer wavelengths since the Keldysh parameters are smaller?

We cannot provide a definite answer to this question in case of solids without further experimental studies. However, our observations indicate the following behaviour. In our experiments, HD occurs primarily in the intermediate intensity regime where MPAT is predominant. Increasing the wavelength or the laser intensity would decrease the Keldysh parameter and the interaction transitions towards the ground-state tunnelling. Experimentally, we observed a decrease in HD at high intensities (see the above figure in which the difference of curves gives HD) most likely due to propagation and other nonlinear effects. At longer wavelengths, these nonlinear effects are expected to occur at higher critical powers, therefore, the HD should be stronger over a certain intensity range. Clear signatures of increase in HD would be better observed when the propagation and nonlinear effects can be minimized as in gas-phase experiments. Our gas-phase results based on differential ionization (not shown, under publication process) does indicate that it is the case.

Reviewer #1 (Remarks to the Author):

The authors response to referee comments are extremely detailed. I think the authors have done a decent job at improving the accessibility of the work to readers by improving the Introduction and definitions of their mechanisms. I believe the work is certainly novel enough to be published in this journal; ultimately the validity and interpretation of the results can be judged by the community.